palaeontology

Odonata, Eocene, geometric morphometrics

**Author for correspondence:**
Eugenia Romero-Lebrón
e-mail: eugeniaromerolebron@gmail.com

# Geometric morphometrics of endophytic oviposition traces of Odonata (Eocene, Argentina)

Eugenia Romero-Lebrón[1], Raquel M. Gleiser[1] and Julián F. Petrulevičius[2]

[1]Centro de Relevamiento y Evaluación de Recursos Agrícolas y Naturales (IMBIV, UNC-CONICET), 5000 Córdoba, Argentina
[2]División Paleozoología Invertebrados, Facultad de Ciencias Naturales y Museo (UNLP), and CONICET, 1900 La Plata, Buenos Aires, Argentina

ER-L, 0000-0002-6518-2189

The insertion of the Odonata ovipositor in the plant tissue generates a scar that surrounds the eggs (trace). In insects, individual egg traces are known to vary in size, but their variation in individual shape is mostly unknown. Twenty-four specimens were obtained from the Laguna del Hunco (Lower Eocene, Chubut) and Río Pichileufú (Middle Eocene, Río Negro), Argentina, which had 1346 oviposition traces (MEF Collection). For the first time, a study of the shape and size of a large number of individual Odonata endophytic egg traces was carried out using traditional (general and mixed linear models) and geometric morphometrics (Fourier elliptical series) to elucidate whether there are changes in size or shape of the individual endophytic egg traces associated with the substrate used at the time of oviposition, if the Lower Eocene traces have varied in relation to those of the Middle Eocene, and if the ichnological classification (*Paleoovoidus arcuatus*, *P. bifurcatus* and *P. rectus*) reflects such variations. We found differences in size ($p < 0.05$), but not in shape, in relation to the variables studied. This could reflect that the shape of Odonata eggs (inferred from the traces), unlike their size, could have a strong evolutionary constraint already observed since the Eocene.

## 1. Introduction

Occasionally, plants serve as substrates for insect oviposition. In fossil leaves, endophytic oviposition is revealed by the presence of scars (traces) on the surface, which is generated in response to the lesion caused by the ovipositor [1–5].

Since the Palaeozoic, there is evidence of endophytic oviposition in fossil leaves [6,7]. The traces are generally oval [5,6,8,9] and are usually arranged on the leaf according to a pattern. Based on the oviposition pattern followed by the traces, these are classified (among others) as *Paleoovoidus rectus* if the traces are arranged along a linear pattern, *Paleoovoidus bifurcatus* if they occur in pairs forming double or V-shaped rows and *Paleoovoidus arcuatus* when they describe a curved or zigzag pattern. The *Paleoovoidus* ichnogenus proposed by Vasilenko [2] is characterized by medium-sized, elongated, narrow, ovoid or lens-shaped structures with a regular arrangement on the leaf blade. These traces, defined by the reaction tissue of the leaf, are narrow at one end, and are often presented as a dark spot (definition proposed by Vasilenko [2] and redefined by Sarzetti *et al*. [3]). At present, more than a dozen species are included in this ichnogenus [10].

Odonatoptera is one of the oldest lineages of winged insects (Pterygota) that reaches today. They are recorded from the Lower Carboniferous (325 Ma) with already four orders [11], and have a relatively rich fossil record [12]. Odonates are predatory insects, with aquatic or semi-aquatic nymphs. Sexually active Odonata gather in or around bodies of water to mate and lay their eggs. They have two oviposition strategies: endophytic and exophytic [12]. In endophytic behaviour eggs are inserted into plant tissues, while exophytic females release eggs in the water or deposit them on the surface of periaquatic objects [13]. After mating, the female with endophytic oviposition selects a plant substrate and lacerates the plant tissue with the cutting shells in its ovipositor to generate a cavity where the egg is inserted. It is the scars on the surface resulting from the insertion of the egg by the ovipositor which typically fossilize.

Insect eggs come in an incredible diversity of shapes and sizes [14–16]. Variations in egg size can be found within different populations (e.g. [17,18]) or even between eggs from the same clutch of a female (e.g. [19]). Egg size also varies temporally and spatially, and may be related to changes in population quality and abundance [20]. Among odonates, species differ in reproductive traits such as egg size, which is an important feature, as it affects the larval size and developmental performance [19]. Corkum *et al*. [21] showed that hatchlings from larger eggs are larger than those from smaller eggs. The size of immature stages is relevant if we consider that in Odonata cannibalism and predation are highly represented, with larger individuals feeding on smaller ones [22–25]. The relative size and shape of the egg are assumed to be related with the shape and size of the trace formed in the substrate. The shape of individual oviposition traces has not been evaluated except by Romero-Lebrón *et al*. [26] who showed that the traces have a range of shapes that are consistent with position differences of the female due to oviposition behaviour. That study was conducted on a leaf of *Eucalyptus chubutensis* (Berry) González (in part) [27] (Myrtaceae) from Laguna del Hunco (Chubut, Argentina) (Early Eocene) that showed traces of individual Odonata eggs that were previously classified to two ichnospecies (*P. arcuatus* and *P. rectus*, [3]), which would have been performed by a single female.

Despite the efforts of Romero-Lebrón *et al*. [26], it is still not clear whether the shape of individual traces varies in relation to factors such as the ichnological classification, the age of the fossil or the taxonomic identification of the substrate. The aim of this study is to carry out a detailed study of the shape and size of individual endophytic oviposition traces of Lower and Middle Eocene fossil odonates in Patagonia, Argentina, and thus to elucidate whether there are changes in shape or size of the individual traces between these two periods, and/or if they are associated with the substrate used at the time of oviposition, and their ichnological classification. This is an unprecedented large-scale study and for this purpose, we have considered the geometric morphometrics (by Fourier elliptical analysis) and the traditional morphometrics of the individual oviposition traces.

## 2. Material and methods

The complete collection of oviposition traces of the Egidio Feruglio Paleontological Museum (MEF), located in Trelew, Province of Chubut, Argentina, was reviewed. We photographed and studied in detail 24 materials that possessed traces of endophytic ovipositions attributed to Odonata, 23 of which are published in Sarzetti *et al*. [3]. The materials have two collection numbers, one palaeobotanical (MPEF-Pb) and one ichnological (MPEF-IC). MPEF-Pb-2216 has two ichnological classifications (MPEF-IC-1376 and MPEF-IC-1392). The analyses were carried out based on ichnological classifications, thus the final number of samples is 25. Sample MEF-IC-1382 was classified as *P. arcuatus* according to Krassilov [28] because the traces follow a curved pattern (electronic supplementary material, appendix S1). The substrate was assigned to a Dicotyledonous leaf. The traces come from the Patagonian Eocene localities of Laguna del Hunco (Ypresian, 52 Ma; [29]) and Río Pichileufú (Lutetian, 48 Ma; [29]) (figure 1), both well-known localities concerning their Odonata [30–38] and plant diversity [29,39–41].

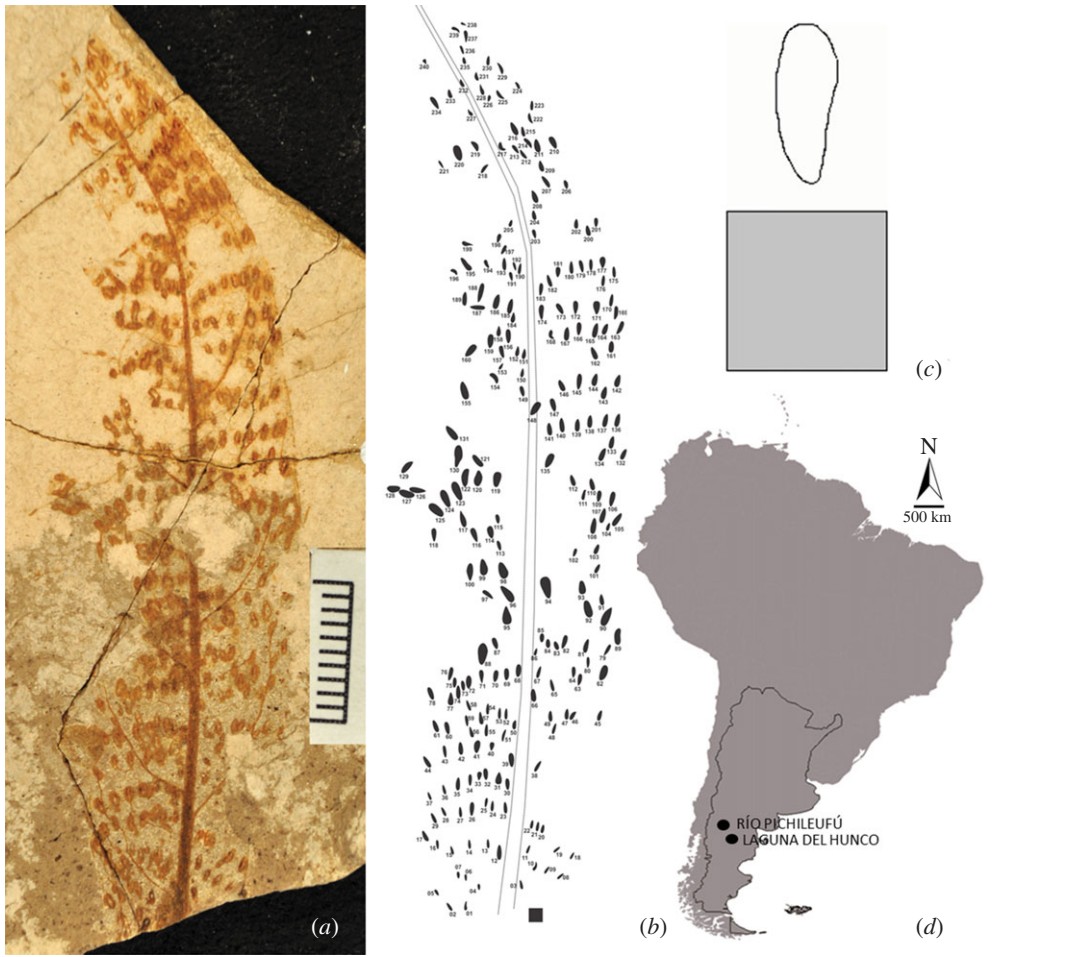

**Figure 1.** (*a*) Image of the Dicotyledonous leaf with traces (*Paleoovoidus arcuatus*) of Odonata endophytic eggs located along the entire leaf surface (MPEF-IC-1388), scale: 1 cm. (*b*) Superposition of layers to the main photograph of the leaf in which the contours of each trace were marked in detail. (*c*) Detail of an egg trace with the scale (square 1 × 1 mm) in the lower margin. (*d*) Map of Argentina showing the localities of Laguna del Hunco (Lower Eocene, Ypresian, 52 Ma) and Río Pichileufú (Middle Eocene, Lutetian, 48 Ma) in Patagonia.

The materials were observed and photographed with a Nikon SMZ1000 magnifier with built-in Nikon DS-Fi1-L2 camera. For this purpose, the leaf fossils containing traces of endophytic ovipositions were placed on a horizontal support, perpendicular to the camera's charged-coupled device (CCD). The distance between the objective of the camera and the sample to be photographed was held constant to avoid distortion of the micro-photographs, which were generated in a 2560 × 1920 pixels size in TIF format. The images were edited so that all the partial photographs completed the entire leaf. The contours of each trace were marked in detail on a digitally superimposed layer (figure 1*b*). This layer was used in the analysis of traditional morphometrics and geometric morphometrics.

Traditional morphometrics is based on measurements of linear distances, such as length and width. Geometric morphometrics, on the other hand, capture the complete geometry of the organism [42]. Although there are numerous mathematical techniques for describing closed contours (for a review of these, see [43–45]), variants of Fourier analysis are generally used. In general, this technique consists of analysing the contribution of the coefficients of a trigonometric function that reproduces a certain curve as accurately as possible. Fourier analysis coefficients then become descriptors of the shape of that curve that can be compared with the corresponding coefficients of other curves using various multivariate statistical techniques.

## 2.1. Traditional morphometrics

The following measurements were taken for each oviposition trace using ImageJ 1.51n: length (measured as the longest part of the trace), width (measured as the widest part of the trace,

perpendicular to length), area and perimeter (table 1). The results were analysed using generalized mixed linear models (GLMM)

$$Y = \mu + \tau \text{ locality/age} + \tau \text{ ichnotaxonomy} + \sigma \text{ substrate } > \sigma \text{ material} + \varepsilon,$$

where $Y$ represents the measures of the traces (length, width, etc.). The model has two fixed factors, the 'age/locality' factor (with two levels, Laguna del Hunco/Lower Eocene, Río Pichileufú/Middle Eocene) and the 'ichnotaxonomy' factor (with three levels, one per ichnotaxon), and two random effects: the taxonomic identity of the oviposition substrate (with 10 levels, depending on the lowest level of taxonomic resolution achieved) and identity of the material (25 levels); the material was nested in the substrate.

On the other hand, the effect of the substrate on the dimensions of the traces was evaluated according to the following model:

$$Y = \mu + \tau \text{ substrate} + \sigma \text{ ichnotaxonomy} + \sigma \text{ locality} + \sigma \text{ material} + \varepsilon,$$

where the substrate was considered a fixed factor and the three random factors included were ichnotaxonomy, locality and material identity.

Considering that the previous model included pseudo replicas for several substrates of which there was only one representative, on the other hand, at the family level only Myrtaceae and Proteaceae were compared, since they were the only ones for which there were repetitions, according to the following model:

$$Y = \mu + \tau \text{ family} + \sigma \text{ material} + \varepsilon,$$

where the family was considered a fixed factor with two levels (Myrtaceae and Proteaceae) and the material was the random factor (all this material comes from Laguna del Hunco and was *P. arcuatus*).

In all the models, the fulfilment of the assumptions of normality and homoscedasticity were verified, and in its defect the correction was carried out by means of the function varIdent. In the case of significant effects, in order to determine which means differed, the least significant difference (LSD) Fisher *a posteriori* test was carried out (Alpha = 0.05).

## 2.2. Geometric morphometrics

For geometric morphometrics, elliptic analysis of Fourier was performed, following the methodology described in Romero-Lebrón *et al*. [26]. We worked with individual images of the contour of each of the traces (figure 1*c*). The free statistical package SHAPE 1.3 [46] was used to calculate Fourier coefficients. Twenty harmonics were taken [47] (tested with a lower number of harmonics; 20 harmonics were better adjusted to the reference shape). With the numerous variables produced, a principal components analysis (PCA) was performed using covariance matrices. For this, the free distribution statistical package PAST 3.15 was used [48]. Finally, PrinPrint (SHAPE 1.3 subprogram) was used to visualize the variation in shape represented by each principal component (PC).

# 3. Results

A total of 1346 traces of Odonata oviposition from 24 materials (25 samples) were studied (traces that were not well defined and had incomplete contours were excluded). Based on the botanical taxonomy of the material (information provided by the MEF except for the sample MEF-IC-1382), a great diversity of substrates used for oviposition is observed. Traces of oviposition were found in 12 Dicotyledons without minor identification, and in seven families: Malvaceae, Myrtaceae, Celtidaceae, Cunoniaceae, Fabaceae, Proteaceae and Sapindaceous (table 1).

Regarding the abundance of substrates used for endophytic oviposition, 50% of the material was only identified at the Dicotyledonous Class level; for the Celtidaceae (*Celtis ameghenoi* Berry), Cunoniaceae (*Cupania latifolioides* Berry), Fabaceae (*Cassia argentinensis* Berry), Malvaceae and Sapindaceous families (*Schmidelia proedulis* Berry) one material of each was found. The Myrtaceae family was the most represented (*Myrcia deltodea* Berry, *Eucalyptus chubutensis* Berry) followed by Proteaceae (*Lomatia occidentalis* Berry) (table 1).

As for the ichnotaxonomic classification already attributed to the reviewed materials, there were three ichnospecies: *P. arcuatus*, *P. bifurcatus* and *P. rectus* (table 1). The ichnospecies *P. arcuatus* was the most frequent ($n = 22$) representing 91.67% of total materials, followed by *P. bifurcatus* and *P. rectus*, each

**Table 1.** Materials analysed: identification, locality (RP, Río Pichileufú; LH, Laguna del Hunco), palaeobotanical classification, ichnotaxonomic classification, number of traces and measurements (average ± s.e.). Values are expressed in mm.

| MPEF-IC | location | substrate | ichnospecies | n traces | length (L) | width (W) | L/W index | area (mm²) | perimeter |
|---|---|---|---|---|---|---|---|---|---|
| 1386 | RP | Dicotyledonous | P. arcuatus | 8 | 1.28 ± 0.04 | 0.77 ± 0.02 | 1.67 ± 0.05 | 0.75 ± 0.03 | 3.47 ± 0.08 |
| 1388 | RP | Dicotyledonous | P. arcuatus | 240 | 0.87 ± 0.01 | 0.33 ± 0.01 | 2.75 ± 0.03 | 0.21 ± 0.01 | 2.02 ± 0.04 |
| 1390 | RP | Dicotyledonous | P. arcuatus | 126 | 1.21 ± 0.01 | 0.46 ± 0.01 | 2.65 ± 0.03 | 0.48 ± 0.01 | 3.09 ± 0.03 |
| 1391 | RP | Dicotyledonous | P. arcuatus | 58 | 0.81 ± 0.02 | 0.29 ± 0.01 | 2.89 ± 0.07 | 0.17 ± 0.01 | 1.90 ± 0.05 |
| 1393 | RP | Dicotyledonous | P. arcuatus | 44 | 1.34 ± 0.02 | 0.55 ± 0.01 | 2.45 ± 0.05 | 0.54 ± 0.02 | 3.32 ± 0.05 |
| 1382 | LH | Dicotyledonous | P. arcuatus | 18 | 0.99 ± 0.04 | 0.41 ± 0.02 | 2.44 ± 0.04 | 0.32 ± 0.03 | 2.45 ± 0.11 |
| 1367 | LH | Dicotyledonous | P. arcuatus | 294 | 0.95 ± 0.01 | 0.35 ± 0.35 | 2.82 ± 0.03 | 0.26 ± 0.01 | 2.31 ± 0.03 |
| 1371 | LH | Dicotyledonous | P. arcuatus | 30 | 0.91 ± 0.02 | 0.31 ± 0.01 | 2.95 ± 0.08 | 0.22 ± 0.01 | 2.19 ± 0.04 |
| 1372 | LH | Dicotyledonous | P. arcuatus | 21 | 1.18 ± 0.02 | 0.52 ± 0.01 | 2.29 ± 0.05 | 0.50 ± 0.02 | 3.05 ± 0.05 |
| 1375 | LH | Dicotyledonous | P. arcuatus | 36 | 1.12 ± 0.03 | 0.38 ± 0.02 | 3.00 ± 0.10 | 0.33 ± 0.02 | 2.68 ± 0.08 |
| 1380 | LH | Dicotyledonous | P. arcuatus | 31 | 0.91 ± 0.03 | 0.33 ± 0.01 | 2.84 ± 0.09 | 0.19 ± 0.01 | 2.09 ± 0.08 |
| 1383 | LH | Dicotyledonous | P. arcuatus | 39 | 1.16 ± 0.04 | 0.40 ± 0.02 | 2.97 ± 0.09 | 0.37 ± 0.03 | 2.78 ± 0.10 |
| 1384 | LH | Dicotyledonous | P. arcuatus | 21 | 1.07 ± 0.05 | 0.47 ± 0.03 | 2.36 ± 0.09 | 0.39 ± 0.04 | 2.66 ± 0.14 |
| 1381 | LH | Malvaceae | P. arcuatus | 5 | 0.79 ± 0.10 | 0.25 ± 0.03 | 3.12 ± 0.14 | 0.15 ± 0.03 | 1.83 ± 0.23 |
| 1370 | LH | Celtidaceae | Celtis ameghenoi | 65 | 1.35 ± 0.02 | 0.47 ± 3 × 10⁻³ | 2.85 ± 0.04 | 0.49 ± 0.01 | 3.26 ± 0.03 |
| 1374 | LH | Cunoniaceae | Cupania latifolioides | 38 | 1.00 ± 0.02 | 0.34 ± 0.01 | 3.06 ± 0.10 | 0.26 ± 0.01 | 2.36 ± 0.04 |
| 1377 | LH | Fabaceae | Cassia argentinensis | 30 | 0.98 ± 0.03 | 0.35 ± 0.01 | 2.79 ± 0.08 | 0.25 ± 0.01 | 2.33 ± 0.06 |
| 1378 | LH | Proteaceae | Lomatia occidentalis | 84 | 1.10 ± 0.01 | 0.42 ± 4 × 10⁻³ | 2.63 ± 0.03 | 0.36 ± 0.01 | 2.71 ± 0.03 |
| 1389 | LH | | Lomatia occidentalis | 49 | 0.77 ± 0.02 | 0.32 ± 0.01 | 2.44 ± 0.06 | 0.17 ± 0.01 | 1.84 ± 0.05 |
| 1385 | LH | Sapindaceous | Schmidelia proedulis | 6 | 1.75 ± 0.13 | 0.62 ± 0.03 | 2.86 ± 0.28 | 0.86 ± 0.08 | 4.27 ± 0.26 |
| 1368 | LH | Myrtaceae | Myrcia deltodea | 46 | 0.84 ± 0.02 | 0.41 ± 0.01 | 2.11 ± 0.06 | 0.24 ± 0.01 | 2.07 ± 0.06 |
| 1369 | LH | | Eucalyptus chubutensis | 4 | 0.79 ± 0.08 | 0.33 ± 0.02 | 2.49 ± 0.37 | 0.19 ± 0.02 | 1.94 ± 0.16 |
| 1373 | LH | | Eucalyptus chubutensis | 28 | 0.84 ± 0.04 | 0.35 ± 0.02 | 2.51 ± 0.10 | 0.22 ± 0.02 | 2.05 ± 0.10 |
| 1376 | LH | | Eucalyptus chubutensis | 9 | 0.80 ± 0.05 | 0.31 ± 0.02 | 2.63 ± 0.19 | 0.15 ± 0.02 | 1.75 ± 0.11 |
| 1392 | LH | | Eucalyptus chubutensis | 16 | 0.84 ± 0.05 | 0.23 ± 0.01 | 3.75 ± 0.20 | 0.12 ± 0.01 | 1.80 ± 0.10 |

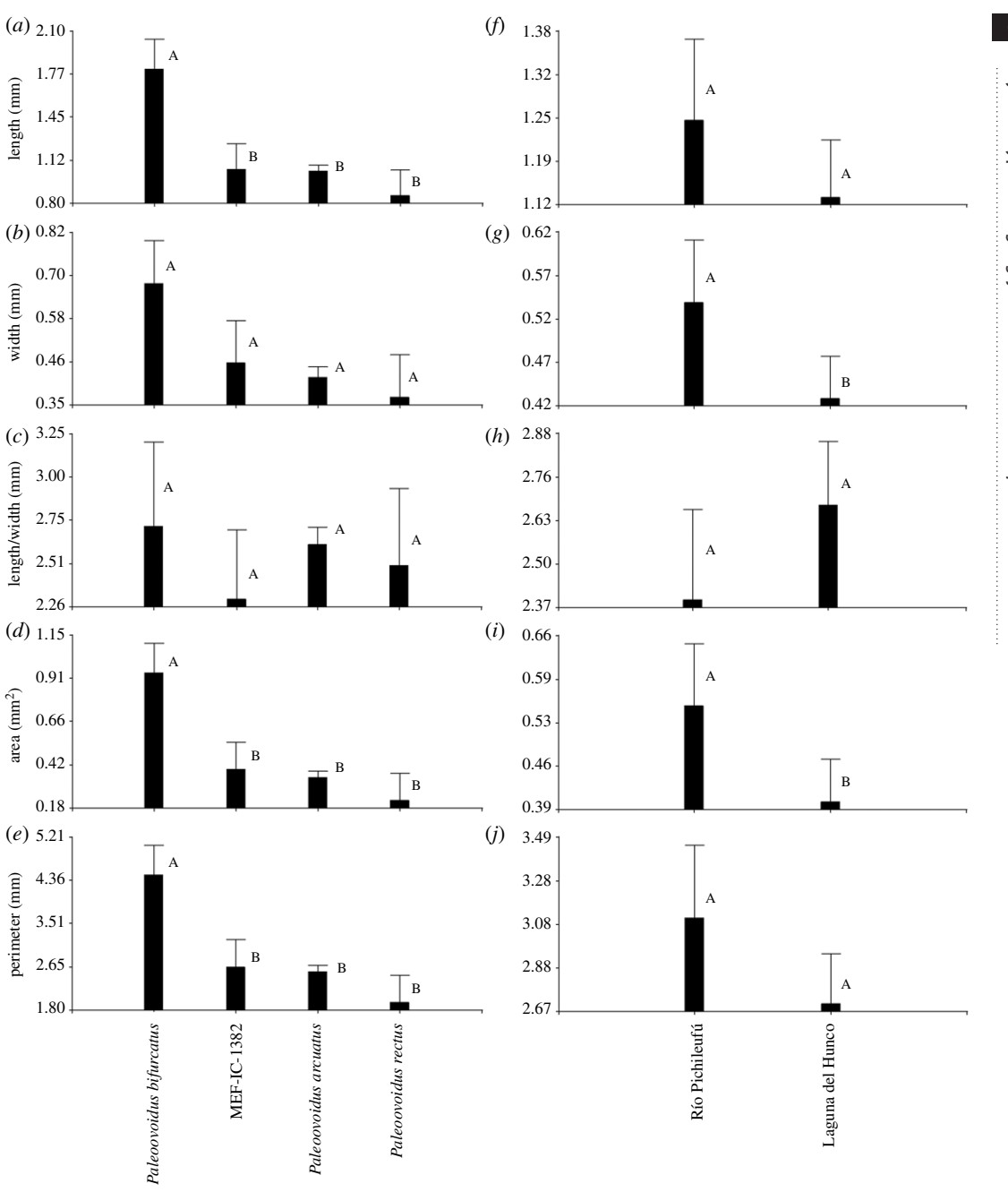

**Figure 2.** Traditional morphometrics of the traces of Odonata endophytic oviposition in relation to their ichnotaxonomy (*a–e*) and for *Paleoovoidus arcuatus* in relation to the age/locality (*f–j*). Mean values + standard errors are displayed. Different letters between columns indicate statistically significant differences (*p* < 0.05).

with only one specimen. As previously stated, we assigned one unidentified material (MPEF-IC-1382) to *P. arcuatus*. *Paleoovoidus arcuatus* is present in a great diversity of substrates. On one occasion, *P. arcuatus* coexists with *P. rectus* in a single specimen of the family Myrtaceae (*E. chubutensis*) (Romero-Lebrón *et al*. [26] analysed this material in detail). The only specimen of *P. bifurcatus* is found in the only sample with traces of the family Sapindaceous (*S. proedulis*) (table 1).

## 3.1. Traditional morphometrics

*Relationship of metrics with provenance (age), ichnotaxonomic classification and oviposition substrate.* Figure 2 shows that the individual oviposition traces from Río Pichileufú (Middle Eocene) were significantly larger (greater length, width, area and perimeter) than those from Laguna del Hunco (Lower Eocene)

(based on GLMM, $p < 0.05$). When comparing by ichnotaxonomic classification, the traces of *P. bifurcatus* (MPEF-IC-1385) hold the greatest length, area and perimeter ($p < 0.05$), while those of *P. arcuatus* and *P. rectus* do not differ from each other ($p < 0.05$). Material MPEF-IC-1382 (which we assigned to *P. arcuatus*) has a trace length similar to other *P. arcuatus* and *P. rectus* (figure 2). The substrate with traces of significantly greater length, area and perimeter is *S. proedulis* (classified as *P. bifurcatus*); while the other substrates in general show statistically non-significant ($p > 0.05$) variations in all dimensions (figure 3). There were also no significant differences in trace dimensions between the families Myrtaceae and Proteaceae.

## 3.2. Geometric morphometrics

For each of the 25 substrate samples, a PCA of their Fourier coefficients was performed to describe the shape variation of the individual traces (electronic supplementary material, appendices S1–S5 and electronic version). Figure 4 illustrates the variation in shape based on Fourier analysis on four substrates according to ichnotaxon. Depending on the substrate, the PC 1 explained 43.33–98.81% of the total variation of the data (table 2), and discriminated the shape of the apex of the trace. The PC 2 explained from 0.73% to 36.32% of the total variation of the data, discriminating the curvature of the trace.

## 3.3. Ichnotaxonomy and geometric morphometrics

The PCA of Fourier coefficients in relation to trace ichnotaxonomy reduced their variability to three PCs which together explain 92.61% of the total shape variation. PC 1, PC 2 and PC 3, respectively, explain 58.06%, 26.83% and 7.72% of the total variation. A great diversity of oviposition trace morphotypes is observed in *P. arcuatus*. However, no particular trace shape was detected for each taxon, as the morphotypes observed in *P. arcuatus* are shared with *P. rectus* and *P. bifurcatus* (figure 5).

## 3.4. Substrate diversity and geometric morphometrics

The first three PCs of the PCA of Fourier coefficients related to substrate diversity together explain 92.66% of the total variance in the shape of the individual oviposition traces. PC 1, PC 2 and PC 3 explain 64.54%, 21.08% and 7.04% of the total variation, respectively. We assessed the 12 materials identified at the family or species level. Those assigned only to the category 'Dicotyledonous' were excluded in this instance of analysis (see table 1 for details of the materials). The Malvaceae family occupies a narrow morphospace that mostly does not overlap with *L. occidentalis*. By contrast, *E. chubutensis* has a wide morphospace that overlaps with all other substrates (figure 6*a*). A closer examination of *E. chubutensis* materials also shows a wide overlap between samples (figure 6*b*), lower for MPEF-IC-1392 that has narrower traces compared to the other three.

## 3.5. Comparison between Lower and Middle Eocene

The first three components of a PCA on the Fourier coefficients considering the ages/localities of the samples explain an accumulated 92.61% of the total variance observed in the shape. PC 1, PC 2 and PC 3 explain 58.06%, 26.83% and 7.72% of the total variation, respectively. A large overlap of morphotypes is observed, regardless of age/locality (figure 7).

# 4. Discussion

## 4.1. Substrate

The choice of substrate and environment that adults choose as a place for the development of their offspring is complex and important, as it will influence the reproductive success of the species [49]. Vegetation composition is considered to be an important indicator for current odonates for habitat selection of future larvae [50,51]. In fossil material, some authors associate substrate for oviposition to certain families of Zygoptera. According to Hellmund & Hellmund [52,53], in the fossil record Lestidae females lay their eggs in specific substrates such as *Daphnogene* leaves (Lauraceae), while Coenagrionidae females do not appear to be selective. Caution must be exercised in assigning substrates to recent families for ovipositions in the Eocene as there are extinct families of Odonata [30], like Frenguelliidae,

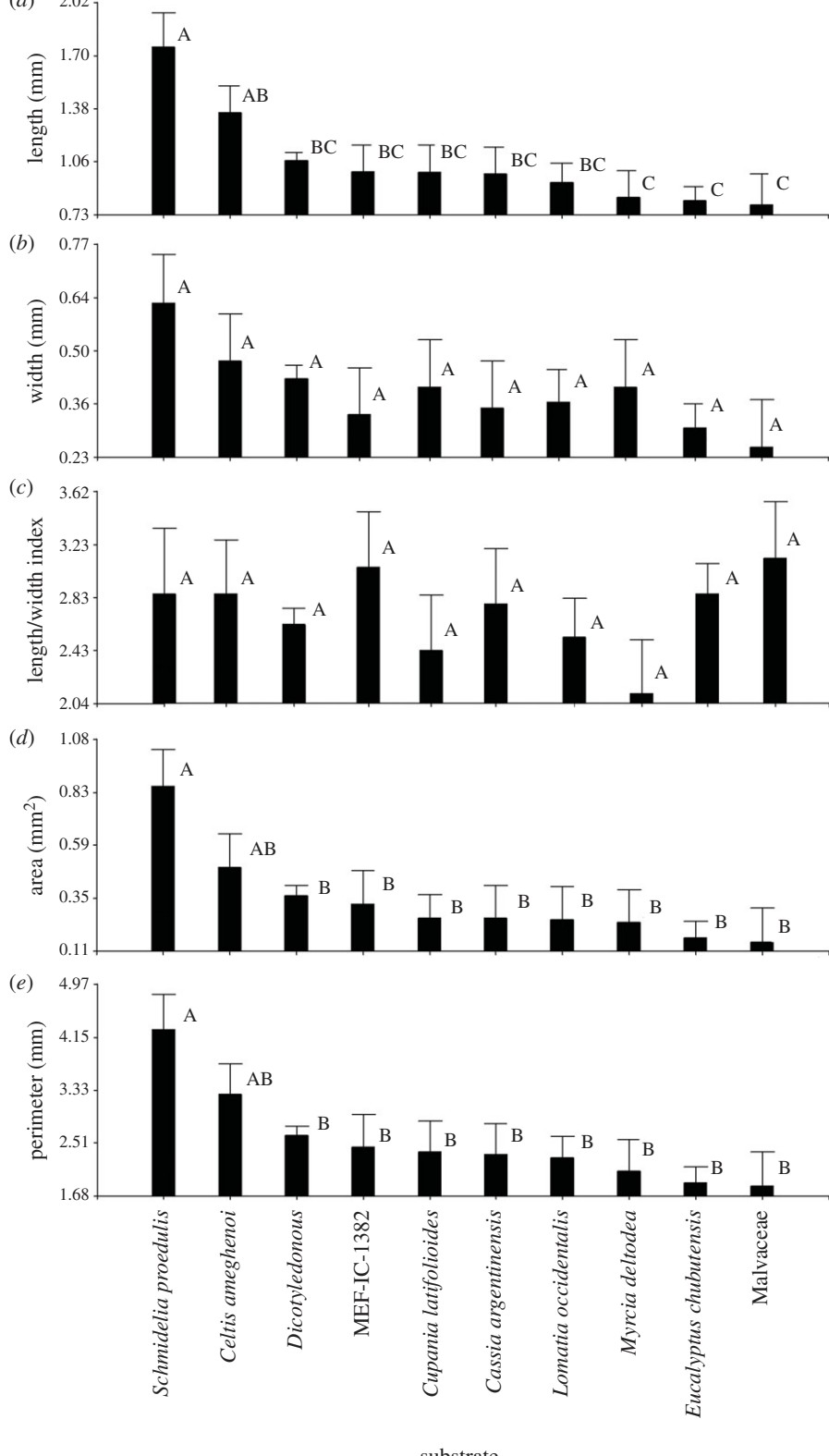

**Figure 3.** Traditional morphometrics of the oviposition traces of Odonata in relation to the substrate used. Average + standard error shown. Different letters between columns indicate statistically significant differences ($p < 0.05$).

Austroperilestidae and Burmagomphidae in Laguna del Hunco and Río Pichileufú [31,32,34,37] that could be also the producers [30]. Concerning a more reliable attribution of ovipositions to a certain genus or family of plants, in our studies, it is difficult to assign a preference for substrate because most plant taxa were represented by one or a few samples. However, all substrates with traces of oviposition analysed

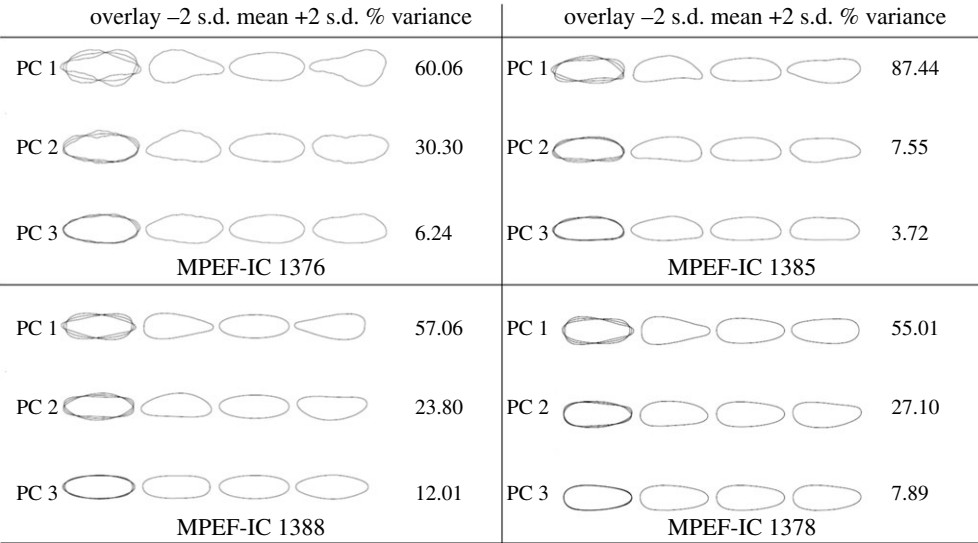

**Figure 4.** Shape variation in the traces of oviposition in fossil leaves from Laguna del Hunco and Río Pichileufú according to ichnotaxon. MPEF-IC 1376 *Paleoovoidus rectus* (Laguna del Hunco); MPEF-IC 1385 *Paleoovoidus bifurcatus* (Laguna del Hunco); MPEF-IC 1388 *Paleoovoidus arcuatus* (Río Pichileufú); MPEF-IC 1378 *Paleoovoidus arcuatus* (Laguna del Hunco). The shape variation is represented in standard deviation units (−2 and +2) and the mean shape is the mean of the Fourier coefficients for all forms analysed.

**Table 2.** Summary table of the principal components (PC) analysis of the Fourier coefficients for each material analysed.

|  | PC 1 | PC 2 | PC 3 |
| --- | --- | --- | --- |
| MPEF-IC 1367 | 56.84 | 30.89 | 6.14 |
| MPEF-IC 1368 | 62.34 | 28.89 | 5.73 |
| MPEF-IC 1369 | 98.81 | 0.73 | 0.46 |
| MPEF-IC 1370 | 54.13 | 20.53 | 11.28 |
| MPEF-IC 1371 | 49.20 | 36.32 | 6.89 |
| MPEF-IC 1372 | 64.53 | 25.45 | 5.04 |
| MPEF-IC 1373 | 67.95 | 10.75 | 9.89 |
| MPEF-IC 1374 | 75.87 | 17.16 | 3.30 |
| MPEF-IC 1375 | 57.94 | 28.70 | 4.85 |
| MPEF-IC 1376 | 60.06 | 30.30 | 6.24 |
| MPEF-IC 1377 | 65.66 | 22.58 | 5.51 |
| MPEF-IC 1378 | 55.01 | 27.10 | 7.89 |
| MPEF-IC 1380 | 43.33 | 27.61 | 10.64 |
| MPEF-IC 1381 | 67.99 | 20.63 | 8.12 |
| MPEF-IC 1382 | 59.41 | 32.41 | 3.32 |
| MPEF-IC 1383 | 59.76 | 23.21 | 10.73 |
| MPEF-IC 1384 | 68.80 | 18.37 | 7.21 |
| MPEF-IC 1385 | 87.44 | 7.55 | 3.72 |
| MPEF-IC 1386 | 68.38 | 24.05 | 3.55 |
| MPEF-IC 1388 | 57.06 | 23.80 | 12.01 |
| MPEF-IC 1389 | 68.48 | 19.25 | 5.12 |
| MPEF-IC 1390 | 65.87 | 16.11 | 8.20 |
| MPEF-IC 1391 | 53.99 | 30.14 | 9.57 |
| MPEF-IC 1392 | 53.59 | 24.79 | 12.20 |
| MPEF-IC 1393 | 49.28 | 20.51 | 14.86 |

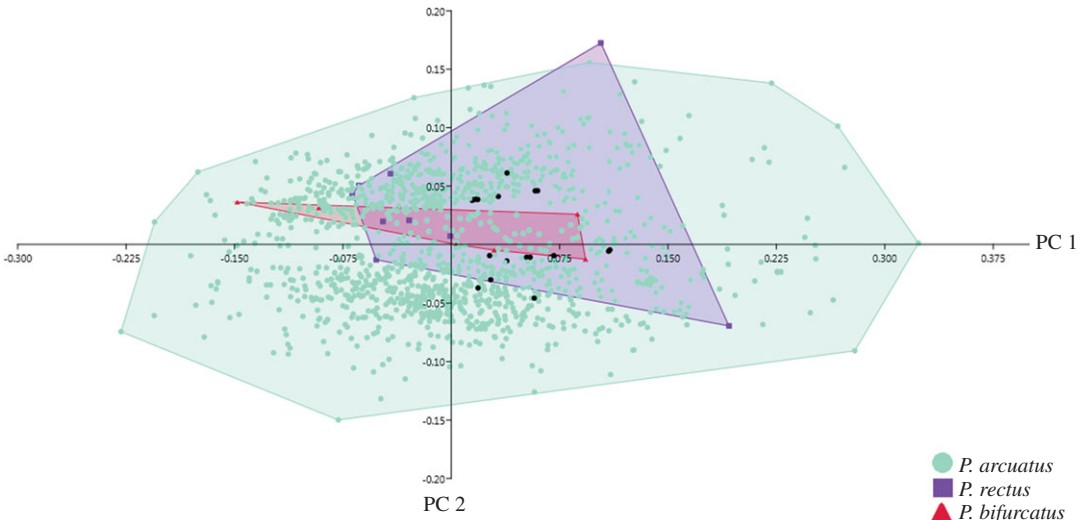

**Figure 5.** Principal components analysis of Fourier coefficients of the individual traces analysed in 25 samples in relation to the three identified ichnotaxa (black dots correspond to MPEF-IC-1382 material assigned to *Paleoovoidus arcuatus*).

(*a*)

(*b*)

**Figure 6.** Principal components analysis of Fourier coefficients of the individual traces analysed in relation to (*a*) 12 oviposition substrates identified at family or species level and (*b*) oviposition traces in *Eucalyptus chubutensis* substrate.

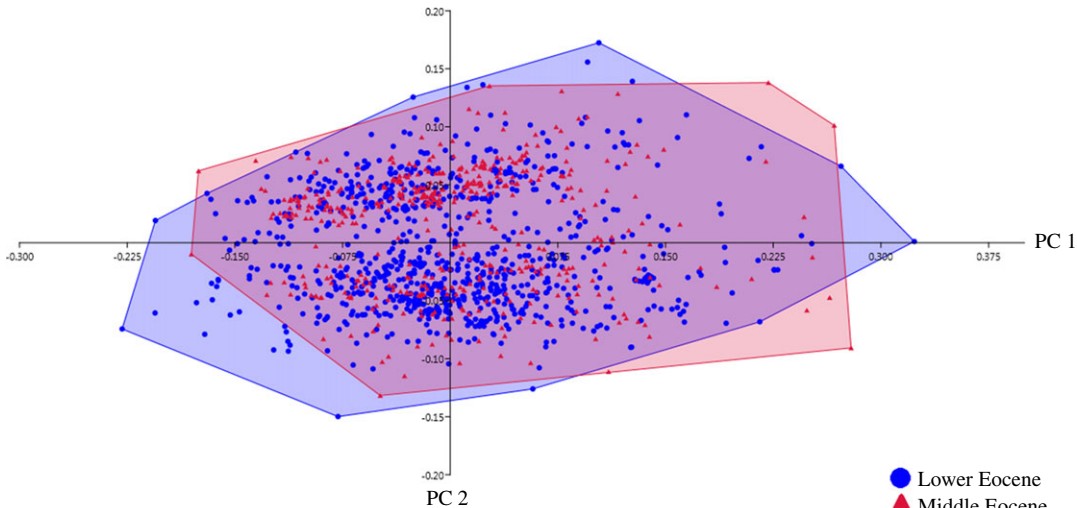

**Figure 7.** Principal components analysis of the Fourier coefficients of the individual traces analysed in 25 samples in relation to their origin: Laguna del Hunco (Lower Eocene, Ypresian, 52 Ma) and Río Pichileufú (Middle Eocene, Lutetian, 48 Ma).

correspond to Dicotyledonea. The most represented family is Myrtaceae, with *E. chubutensis* (Myrtaceae) and *L. occidentalis* (Proteaceae) being the most abundant species. Wilf *et al.* [39,40] analysed in detail the diversity of the flora of Laguna del Hunco and Río Pichileufú based on more than 3500 fossil materials, providing a list of the 30 most abundant plant species of Laguna del Hunco. The substrates in which traces were detected belong to taxa of plants included in this list; therefore, the abundance of substrates chosen for oviposition agrees with their relative abundance in the environment, then there would be no evidence of a pressure of selection of substrate. On the other hand, *M. deltodea* is not one of the 30 most frequent plants, opening the question as to whether it reflects a selection of this substrate.

## 4.2. Traditional morphometrics

Historically, linear measurements have been widely used to describe fossil structures. Sarzetti *et al.* [3] provide measurements of some of the materials analysed in this study that differ from our estimates. The differences between Sarzetti *et al.* [3] and our measurements could be due in part to the fact that they considered different numbers of traces. For example, in the material identified as *P. rectus* (MPEF-IC-1376) they considered 12 traces, while we identified 13. However, four of them were excluded from the analyses because they were incomplete. In the case of *P. arcuatus*, the differences could be due to the fact that they do not consider the material MPEF-IC-1382, but instead, material USNM 40389 belonging to the National Museum of Natural History, Smithsonian Institution (Washington, DC) not evaluated by us. In the case of *P. bifurcatus*, the differences observed could be due to different criteria when measuring the trace. In fact, differences of criteria in the measurement are a frequent problem for those who work with linear measurements that could be reduced using the technique of geometric morphometrics.

## 4.3. Ichnotaxonomy

*Paleoovoidus arcuatus* is the most abundant ichnotaxon in Laguna del Hunco and Río Pichileufú. The ichnogenus *Paleoovoidus* Vasilenko [2] was originally described for traces of oviposition belonging to the Upper Jurassic–Lower Cretaceous of Russia. Since then it is presented in great abundance, mainly in the Palaeogene and Neogene (manuscript in preparation). Authors such as Sarzetti *et al.* [3], and Petrulevičius [30] and colleagues [4], cite *P. arcuatus* in the Eocene and Oligocene, but there could be numerous synonyms if we consider only the spatial arrangement of the set of traces (e.g. 'Coenagrionidae Type' or DT54 and DT 100) that would identify it from the Permian (e.g. [54]) and even from the Carboniferous (e.g. [6]). On one occasion *P. arcuatus* and *P. rectus* are present in the same leaf (MPEF-Pb-2216 [MPEF-IC-1376 and MPEF-IC-1392]), being interpreted as traces made by the same female [26].

Of the three ichnotaxa present, traces of *P. bifurcatus* were the longest. This ichnospecies is mentioned for the first time in Sarzetti *et al.* [3]. Oviposition traces of insects with a pattern similar to *P. bifurcatus* are found in the Lower Permian of India [55], but the size of the traces is considerably smaller (0.1–0.5 mm). Later,

Hellmund & Hellmund [52,53,56] and Petrulevičius *et al.* [4] report Odonata trace eggs in Upper Oligocene materials from Germany with this type of spatial arrangement, also in smaller size (0.5–0.9 mm). Considering only the spatial arrangement of the traces of insect oviposition, in the 'curved', 'straight' and 'bifurcated' arrangement, the 'bifurcated' is recorded from Permian to Oligocene (except Palaeocene) (manuscript in preparation). On the other hand, traces of oviposition of insects with a similar length range are recorded in the Permian [57–60], Triassic [10,61], Jurassic [62], Cretaceous [63], Palaeocene [64], Oligocene [53,65,66] and Miocene [56], but in none of these cases do the traces have spatial arrangement similar to *P. bifurcatus*. Therefore, the characteristics of this ichnospecies could be unique to Argentinian Patagonia, although we would need more samples to reinforce this hypothesis.

The number of oviposition traces per leaf in the studied materials is very variable, with a minimum–maximum range of 5–294 traces. In the bibliography, from Cretaceous onwards the number of endophytic ovipositions traces per leaf increases considerably (manuscript in preparation). Krassilov *et al.* [67] cite for the Lower Cretaceous of Israel, 250 traces of oviposition of Zygoptera in a Dicotyledonous leaf, and from that moment on, it is not strange to find high numbers of traces, like those found in our study.

## 4.4. Geometric morphometrics

While numerous papers have contributed trace data, most descriptions of traces of fossil ovipositions have been qualitative, based on linear dimensions (length–width). For the first time, a large number of Odonata's endophytic oviposition traces are analysed with geometric morphometrics, except for the Romero-Lebrón *et al.* [26] observation, but their analysis was done on a single leaf. Using geometric morphometrics we observed a great diversity of oviposition trace morphotypes in the ichnospecies *P. arcuatus*. These morphotypes are shared with *P. rectus* and *P. bifurcatus*. The diversity of morphotypes of *P. arcuatus* could be due to the great abundance of identified materials of this ichnotaxon, since of *P. rectus* and *P. bifurcatus* there is only one representative of each. The greatest variation in the shape of the oviposition traces corresponded to the shape close to the apex, and to a lesser extent to the curvature and convexity of the trace. These three variations are also observed in the morphology of extant Zygoptera eggs (manuscript in preparation).

On the other hand, no particular morphotype was observed for each oviposition substrate. Regardless of the substrate, the average shape of the traces does not vary. Even though Malvaceae showed a narrow morphospace that was separated from that of *L. occidentalis*, it is not clear whether this is an artefact due to the relatively low number of traces on the first substrate. By contrast, *E. chubutensis* showed the widest morphospace, which in part could be explained by the larger number of separate samples ($N = 4$) analysed. However, *M. deltodea* also shows a wide morphospace, with all traces measured from the same substrate material. Therefore, we interpret that the substrate would not be exerting differential shape modification pressures on the oviposition traces between the studied materials. In addition, a marked overlap was observed in the shape of the traces regardless of their locality of origin (LH—Lower Eocene—and RP—Middle Eocene), so it could be inferred that the shape of the individual traces would not depend on provenance or age. Therefore, the variations in the shape of the individual traces would not be related to the ichnotaxonomy, substrate used or locality of provenance or age.

In summary, the shapes of Odonata's individual oviposition traces are observed to be stable over 4 Myr (Lower Eocene–Middle Eocene), while individual size varies. Insect eggs exhibit a great diversity of sizes. These variations are observed within populations [17,18] and there are even variations in the eggs that the same female oviposes (e.g. [19]). Therefore, egg size results in a highly variable parameter that is also observed in the traces of Patagonian Odonata (Lower–Middle Eocene). On the other hand, in this work no consistent changes in oviposition trace shape are observed despite the different substrates used, the 4 Myr considered in this study, and this is not reflected by the icnotaxonomy either. According to Legay [15] insects have existed for hundreds of millions of years, but despite their great diversification, the shape of their eggs shows great stability. This is the first work that analyses the change of shape of Odonata's oviposition traces in a scale of 4 Myr and therefore, it would be the first work that empirically reflects the stability of shape proposed by Legay [15]. This could reflect that the shape of Odonata eggs, unlike their size, could have a strong evolutionary constraint already observed since the Patagonian Eocene.

Data accessibility. Our data are deposited at Dryad Digital Repository: https://dx.doi.org/10.5061/dryad.d51c5b015 [68].
Authors' contributions. E.R.-L. took the photographs, did the analysis, interpretation and drafted the first version of the manuscript with intellectual input from all other authors. All authors contributed intellectually critical content and revisions. All authors gave final approval for the final version.

Competing interests. We declare we have no competing interests.

Funding. We received no funding for this study.

Acknowledgements. Thanks are due to Rubén Cúneo and Eduardo 'Dudu' Ruigómez (Egidio Feruglio Museum, Trelew, Argentina) for their help and providing laboratory infrastructure during the visit to the MEF, and Peter Wilf. We thank Claudia Tambussi, Silvia Gnaedinger and Alberto Rodrigues-Capítulo for helpful comments at an early stage of this study. We are grateful to Alejandro Barbeito for his help in editing the images. We appreciate constructive comments from three anonymous reviewers. Funding support for fieldtrip came from grants: PIP 0834 from the National Research Council of Argentina (CONICET); SECyT N875 from the National University of La Plata (UNLP), PICT-2016–4297 from the National Agency of Scientific and Technological Promotion of Argentina (ANPCyT); and DEB-1556666 from the National Science Foundation of USA (NSF).

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
