## [Reviewer comments · Royal Society Open Science]

Review History

RSOS-201126.R0 (Original submission)

Review form: Reviewer 1 (Enrique Peñalver)

Is the manuscript scientifically sound in its present form?

Yes

Are the interpretations and conclusions justified by the results?

Yes

Is the language acceptable?

No

Do you have any ethical concerns with this paper?

No

Have you any concerns about statistical analyses in this paper?

No

Recommendation?

Accept with minor revision (please list in comments)

Comments to the Author(s)

I included some corrections and comments in the pdf file of the manuscript (Appendix A).

Review form: Reviewer 2

Is the manuscript scientifically sound in its present form?

Yes

Are the interpretations and conclusions justified by the results?

Yes

Is the language acceptable?

Yes

Do you have any ethical concerns with this paper?

No

Have you any concerns about statistical analyses in this paper?

Yes

Recommendation?

Accept with minor revision (please list in comments)

Comments to the Author(s)

RSOS-201126

GEOMETRIC MORPHOMETRICS OF ENDOPHYTIC OVIPOSITION TRACES OF ODONATA (EOCENE, ARGENTINA)

The authors have attempted to study the shape variation of the egg traces of Odonata from the Eocene in Argentina, the article sound very original and the use of elliptical analyses of Fourier are indeed interesting, I missed a more sophisticated analysis implemented in the package "Momocs" for Fourier analyses, the use of old implementation is not bad but always can be better.

I have some comments written in the manuscript PDF (Appendix B) please take a look all of them mostly are of form, but I have some principal concern regarding the substrate variation, the authors said very explicit that there are no pattern related, but I noticed at least 3 pattern of 3 different substrate in the PCA, from Eucalyptus chubutensis (very clear variation in mostly all the morphospace) and in Malvaceae (very narrowed group of specimens not superposed to the others).

Nevertheless, I like the idea and I'm sure that the article can be accepted after minor revision comments of the PDF and improve the manuscript

Review form: Reviewer 3

Is the manuscript scientifically sound in its present form?

Yes

Are the interpretations and conclusions justified by the results?

Yes

Is the language acceptable?

Yes

Do you have any ethical concerns with this paper?

No

Have you any concerns about statistical analyses in this paper?

No

Recommendation?

Accept as is

Comments to the Author(s)

Dear Authors,

Congrats for your manuscript. There is tiny mistakes to be corrected with english and one sentence in the discussion that seems awkward to me.

Otherwise, manuscript deserves to be publish as is.

Illustration and the effort of represent each oviposition per leaf in a draw are very good.

Decision letter (RSOS-201126.R0)

Dear Dr Romero-Lebrón

On behalf of the Editors, we are pleased to inform you that your Manuscript RSOS-201126 "GEOMETRIC MORPHOMETRICS OF ENDOPHYTIC OVIPOSITION TRACES OF ODONATA (EOCENE, ARGENTINA)" has been accepted for publication in Royal Society Open Science subject to minor revision in accordance with the referees' reports. Please find the referees' comments along with any feedback from the Editors below my signature.

Please submit your revised manuscript and required files (see below) no later than 7 days from today's (ie 19-Oct-2020) date. Note: the ScholarOne system will 'lock' if submission of the revision is attempted 7 or more days after the deadline. If you do not think you will be able to meet this deadline please contact the editorial office immediately.

Best regards,

on behalf of Dr Jeffrey Thompson (Associate Editor) and Kevin Padian (Subject Editor)
 openscience@royalsociety.org

Associate Editor Comments to Author (Dr Jeffrey Thompson):

Dear Eugenia et al,

Your manuscript has been reviewed by three expert reviewers, and I am happy to say that they have all been very positive about your manuscript. I am thus recommending acceptance with minor revisions. There are a few cases where things are not clear in the manuscript, where things are misspelled, or where the the English could be improved, which has been highlighted by the reviewers in their attached, annotated, files. There are also a number of stylistic errors found in the references. I thus suggest you go through the manuscript with a fine-toothed-comb to correct these errors in spelling, formatting, and clarity prior to resubmission. Please address all suggestions and comments made by the reviewers prior to resubmission.

All the best,
 Jeff Thompson

Reviewer comments to Author:

Reviewer: 1
 Comments to the Author(s)

I included some corrections and comments in the pdf file of the manuscript (attached)

Reviewer: 2
 Comments to the Author(s)

RSOS-201126

GEOMETRIC MORPHOMETRICS OF ENDOPHYTIC OVIPOSITION TRACES OF ODONATA (EOCENE, ARGENTINA)

The authors have attempted to study the shape variation of the egg traces of Odonata from the Eocene in Argentina, the article sound very original and the use of elliptical analyses of Fourier are indeed interesting, I missed a more sophisticated analysis implemented in the package "Momocs" for Fourier analyses, the use of old implementation is not bad but always can be better.

I have some comments written in the manuscript PDF please take a look all of them mostly are of form, but I have some principal concern regarding the substrate variation, the authors said very explicit that there are no pattern related, but I noticed at least 3 pattern of 3 different substrate in the PCA, from Eucalyptus chubutensis (very clear variation in mostly all the morphospace) and in Malvaceae (very narrowed group of specimens not superposed to the others).

Nevertheless, I like the idea and I'm sure that the article can be accepted after minor revision comments of the PDF and improve the manuscript

Reviewer: 3
Comments to the Author(s)

Dear Authors,

Congrats for your manuscript. There is tiny mistakes to be corrected with english and one sentence in the discussion that seems awkward to me.

Otherwise, manuscript deserves to be publish as is.

Illustration and the effort of represent each oviposition per leaf in a draw are very good.

===PREPARING YOUR MANUSCRIPT===

Your revised paper should include the changes requested by the referees and Editors of your manuscript. You should provide two versions of this manuscript and both versions must be provided in an editable format:
one version identifying all the changes that have been made (for instance, in coloured highlight, in bold text, or tracked changes);
a 'clean' version of the new manuscript that incorporates the changes made, but does not highlight them. This version will be used for typesetting.
Please ensure that any equations included in the paper are editable text and not embedded images.

===PREPARING YOUR REVISION IN SCHOLARONE===

-- Ensure that your data access statement meets the requirements at <https://royalsociety.org/journals/authors/author-guidelines/#data>. You should ensure that you cite the dataset in your reference list. If you have deposited data etc in the Dryad repository, please only include the 'For publication' link at this stage. You should remove the 'For review' link.

Author's Response to Decision Letter for (RSOS-201126.R0)

See Appendix C.

Decision letter (RSOS-201126.R1)

Dear Dr Romero-Lebrón,

It is a pleasure to accept your manuscript entitled "GEOMETRIC MORPHOMETRICS OF ENDOPHYTIC OVIPOSITION TRACES OF ODONATA (EOCENE, ARGENTINA)" in its current form for publication in Royal Society Open Science.

on behalf of Mr Jeffrey Thompson (Associate Editor) and Kevin Padian (Subject Editor)
openscience@royalsociety.org

Appendix A**ROYAL SOCIETY
OPEN SCIENCE****GEOMETRIC MORPHOMETRICS OF ENDOPHYTIC
OVIPOSITION TRACES OF ODONATA (EOCENE, ARGENTINA)**

Journal:	Royal Society Open Science
Manuscript ID	RSOS-201126
Article Type:	Research
Date Submitted by the Author:	21-Aug-2020
Complete List of Authors:	Romero-Lebrón, Eugenia; IMBIV GLEISER, RAQUEL M.; IMBIV PETRULEVIČIUS, JULIÁN F.; Facultad de Ciencias Naturales; CONICET
Subject:	palaeontology < BIOLOGY
Keywords:	ENDOPHYTIC, OVIPOSITION, TRACES, ODONATA, GEOMETRIC MORPHOMETRICS
Subject Category:	Organismal and Evolutionary Biology

Author-supplied statements

Relevant information will appear here if provided.

Ethics

Does your article include research that required ethical approval or permits?:

This article does not present research with ethical considerations

Statement (if applicable):

CUST_IF_YES_ETHICS :No data available.

Data

It is a condition of publication that data, code and materials supporting your paper are made publicly available. Does your paper present new data?:

Yes

Statement (if applicable):

Our data are deposited at Dryad: <https://doi.org/10.5061/dryad.d51c5b015>.

Dryad review URL

<https://datadryad.org/stash/share/VFMIABxwz5HFORoIcfipntAxkOMQtX0dUuC5319sJJ4>.

Conflict of interest

I/We declare we have no competing interests

Statement (if applicable):

CUST_STATE_CONFLICT :No data available.

Authors' contributions

This paper has multiple authors and our individual contributions were as below

Statement (if applicable):

[revised manuscript text omitted]

In all the models the fulfillment of the assumptions of normality and
homocedasticity was verified, and in its defect the correction was carried out by
means of the function varident. In case of significant effects, in order to determine
which means differed, the LSD Fisher *a posteriori* test was carried out (Alfa = 0.05).

*Geometric morphometrics*

For geometric morphometric Elliptic analysis of Fourier was performed,
following the methodology described in Romero-Lebrón (26). We worked with
individual images of the contour of each of the traces (Fig. 1 C). The free statistical
package SHAPE 1.3 (45) was used to calculate Fourier coefficients. Twenty
harmonics were taken (46) (tested with a lower number of harmonics; 20 harmonics
were better adjusted to the reference shape). With the numerous variables produced,
a principal component analysis (PCA) was performed using covariance matrices. For

this, the free distribution statistical package PAST 3.15 was used (47). Finally,
PrinPrint (SHAPE 1.3 subprogram) was used to visualize the variation in shape
represented by each principal component.

**RESULTS**

A total of 1346 traces of Odonata oviposition from 24 materials were studied (traces
that were not well defined and had incomplete contours were excluded). As MPEF-
Pb-2216 has two ichnospecies in the same leaf (MPEF-IC-1376 and MPEF-IC-1392)
and the analyses were carried out based on ichnological classifications, the final
number of samples is 25.

The sample MEF-IC-1382, which was originally unidentified, was classified by
153 us as *P. arcuatus* according to Krassilov (48) because the traces follow a curved
pattern (Appendix S1). The substrate was assigned to a dicotyledonous leaf.

Based on the botanical taxonomy of the material (information provided by the
MEF except for the sample MEF-IC-1382), a great diversity of substrates used for
oviposition is observed. Traces of oviposition were found in 12 Dicotyledons without
minor identification, and in 7 families: Malvaceae, Myrtaceae, Celtidaceae,
Cunoniaceae, Fabaceae, Proteaceae, and Sapindaceous (Table 1).

Regarding the abundance of substrates used for endophytic oviposition, 50 %
of the material was only identified at the Dicotyledonous Class level; for the
Celtidaceae (*Celtis ameghenoi* Berry), Cunoniaceae (*Cupania latifolioides* Berry),
Fabaceae (*Cassia argentinensis* Berry), Malvaceae and Sapindaceous families
(*Schmidelia proedulis* Berry) 1 material of each was found. The Myrtaceae family was
the most represented (*Myrcia deltodea* Berry, *Eucalyptus chubutensis* Berry) followed
by Proteaceae (*Lomatia occidentalis* Berry) (Table 1).

As for the ichnotaxonomic classification already attributed to the reviewed
materials, there were three ichnospecies: *Paleoovoidus arcuatus*, *Paleoovoidus*
*bifurcatus* and *Paleoovoidus rectus* (Table 1). The ichnospecies *P. arcuatus* was the

most frequent (n = 22) representing 91.67 % of total materials, followed by *P.*
*bifurcatus* and *P. rectus*, each with only 1 specimen, and one unidentified material
now assigned to *P. arcuatus*. *Paleoovoidus arcuatus* is present in a great diversity of
substrates. On one occasion, *P. arcuatus* coexists with *P. rectus* in a single specimen
of the family Myrtaceae (*Eucalyptus chubutensis*) [Romero-Lebrón (26) analyse this
material in detail]. The only specimen of *P. bifurcatus* is found in the only sample with
traces of the family Sapindaceae (*Schmidelia proedulis*) (Table 1).

*Classical Morphometrics*

*Relationship of metrics with provenance (age), ichnotaxonomic classification and*
*oviposition substrate.* The traces from Río Pichileufú (Middle Eocene) were
significantly wider and of larger area than those from Laguna del Hunco (Lower
Eocene) ($p < 0.05$). The traces of greater length, area and perimeter are those of *P.*
*bifurcatus* (MPEF-IC-1385: 1.75 ± 0.13 mm), while those of *P. arcuatus* and *P. rectus*
do not differ from each other ($p < 0.05$). Material MPEF-IC-1382 (which we assigned
to *P. arcuatus*) has a trace length similar to *P. arcuatus* and *P. rectus* (Fig. 2). The
substrate with traces of greater ($p < 0.05$) length, area and perimeter is *Schmidelia*
*proedulis* (classified as *P. bifurcatus*); while the other substrates in general show
relatively little variation in all dimensions (Fig. 3). There were also no significant
differences in trace dimensions between the families Myrtaceae and Proteaceae.

*Geometric Morphometrics*

For each of the 25 samples a principal component analysis (PCA) of their Fourier
coefficients was performed to describe the shape of the individual traces (Fig. 4.
Appendix S2-S5). The principal component 1 explained between 43.33 % and 98.81
196 % of the total variation of the data (Table 2). In 84 % of the material the shape close
to the apex was discriminated, while in 12 % the apex shape was discriminated. The

198 principal component 2 explained between 0.73 % and 36.32 % of the total variation of
199 the data, discriminating the curvature of the trace.

*Ichnotaxonomy and geometric morphometrics*

The PCA of Fourier coefficients in relation to trace ichnotaxonomy reduced their
variability to three principal components (PC) which together explain 92.61 % of the
total variation observed in the shape. PC 1, PC 2 and PC 3 respectively explain 58.06
205 %, 26.83 % and 7.72 % of the total variation. A great diversity of oviposition trace
morphotypes is observed in *P. arcuatus*. However, no particular trace shape was
detected for each taxon, as the morphotypes observed in *P. arcuatus* are shared with
*P. rectus* and *P. bifurcatus* (Fig. 5).

*Substrate diversity and geometric morphometrics*

The first three principal components of the PCA of Fourier coefficients in relation to
substrate diversity together explain 92.66 % of the total variance in the shape. PC 1,
2 and 3 explain 64.54 %, 21.08 % and 7.04 % of the total variation, respectively. We
assessed the 12 materials identified at family or species level. Those assigned only to
the category "Dicotyledonous" were excluded in this instance of analysis (see Table 1
for details of the materials). No particular morphotype is observed for each substrate
(Fig. 6).

*Comparison between Lower and Middle Eocene*

The analysis of principal components on the Fourier coefficients considering the
ages/localities of the samples reduced the variability to three principal components
(PC) that together capture 92.61 % of the total variance observed in the shape. PC 1,
2 and 3 explain 58.06 %, 26.83 % and 7.72 % of the total variation, respectively. A
large overlap of morphotypes is observed, regardless of age/locality (Fig. 7).

2254
**DISCUSSION**6
*Substrate*

The choice of substrate and environment that adults choose as a place for the
development of their offspring is complex and important, as it will influence the
reproductive success of the species (49). Vegetation composition is considered to be
an important indicator for current odonates for habitat selection of future larvae (50,
51). In fossil material, there are some authors assigning substrate preferences for
oviposition to certain families of Zygoptera. According to Hellmund & Hellmund (52,
53) *Lestidae* females often show substrate preference for certain plants such as
*Daphnogene* leaves (Lauraceae), while *Coenagrionidae* females, however, do not
appear to prefer a plant type in the fossil record. In this respect, we have to be careful
to make attributions to Recent families for ovipositions in the Eocene as there are
extinct families of Odonata (29), like *Frenguelliidae*, *Austroperilestidae* and
*Burmagomphidae* in Laguna del Hunco and Río Pichileufú (30, 31, 33, 36) that could
be also the producers (29). Concerning a more reliable attribution of ovipositions to a
certain genus or family of plants, in our studies, it is difficult to assign a preference for
substrate because most plant taxa were represented by one or a few samples.
However, all substrates with traces of oviposition analysed correspond to
Dicotyledonea. The most represented family is *Myrtaceae*, being *Eucalyptus*
*chubutensis* (*Myrtaceae*) and *Lomatia occidentalis* (*Proteaceae*) the most abundant
species. Wilf (38, 39) analysed in detail the diversity of the flora of Laguna del Hunco
and Río Pichileufú based on more than 3500 fossil materials, providing a list of the
thirty most abundant plant species of Laguna del Hunco. The substrates in which
traces were detected belong to taxa of plants included in this list, therefore, the
abundance of substrates chosen for oviposition, agrees with the relative abundance
in the environment, then there would be no evidence of a pressure of selection of
substrate. On the other hand, *Myrcia deltodea* is not one of the 30 most frequent
plants, opening the question as to whether it reflects a selection by this substrate.

*Classical Morphometrics*

Historically, linear measurements have been widely used to describe fossil structures.
Sarzetti (3) provide measurements of some of the materials analysed in this study
that differ from our estimates. The differences between Sarzetti (3) and our
measurements could be due in part to the fact that they considered different numbers
of traces, for example, in the material identified as *P. rectus* (MPEF-IC-1376) they
considered twelve traces, while we identified thirteen. However, four of them were
excluded from the analyses because they were incomplete. In the case of *P.*
*arcuatus*, the differences could be due to the fact that they do not consider the
material MPEF-IC-1382, but instead material (USNM 40389) belonging to the
National Museum of Natural History, Smithsonian Institution (Washington, DC) not
evaluated by us. In the case of *P. bifurcatus*, the differences observed could be due
to different criteria when measuring the trace. In fact, differences of criteria in the
measurement are a frequent problem for those who work with linear measurements
that could be reduced using the technique of geometric morphometric.

*Ichnotaxonomy*

*Paleoovoidus arcuatus* is the ichnotaxa most abundant in Laguna del Hunco and Rio
Pichileufú. The ichnogenus *Paleoovoidus* Vasilenko (2) was originally described for
traces of oviposition belonging to the Upper Jurassic - Lower Cretaceous of Russia.
Since then it is presented in great abundance, mainly in the Paleogene and Neogene
(manuscript in preparation). Authors such as Sarzetti (3) and Petrulevičius (4, 29),
cite *P. arcuatus* in the Eocene and Oligocene, but there could be numerous
synonyms if we consider only the spatial arrangement of the set of traces (e.g.
"Coenagrionidae Type" or DT54 and DT 100) that would identify it from the Permian
[e.g. (54)] and even from the Carboniferous [e.g. (6)]. On one occasion *P. arcuatus*
and *P. rectus* were presented in the same leaf [MPEF-Pb-2216 (MPEF-IC-1376 and
MPEF-IC-1392)], being interpreted as traces made by one female (26).

Of the three ichnotaxa present, traces of *P. bifurcatus* were the longest. This
ichnospecies is mentioned for the first time in Sarzetti (3). Oviposition traces of insects
with a pattern similar to *P. bifurcatus* are found in the Lower Permian of India (55), but
the size of the traces is considerably smaller (0.1 to 0.5 mm). Later, Hellmund &
Hellmund (52, 53, 56) and Petrulevičius (4) report traces of Odonata oviposition in
Upper Oligocene materials from Germany with this type of spatial arrangement, also
in smaller size (0.5 to 0.9 mm). Considering only the spatial arrangement of the traces
of insect oviposition, in the "Curved", "Straight" and "Bifurcated" arrangement, the
"Bifurcated" is recorded from Permian to Oligocene (except Paleocene) (manuscript
in preparation). On the other hand, traces of oviposition of insects with a similar
length range are recorded in the Permian (57-60), Triassic (10, 61), Jurassic (62),
Cretaceous (63), Paleocene (64), Oligocene (53, 65, 66) and Miocene (56), but in
none of these cases do the traces have spatial arrangement similar to *P. bifurcatus*.
Therefore, the characteristics of this ichnospecies could be unique to Argentinian
Patagonia, although we would need more samples to reinforce this hypothesis.

The number of oviposition traces per leaf in the studied materials is very
variable, with a minimum - maximum range of 5 to 294 traces. In the bibliography,
from Cretaceous onwards the number of endophytic ovipositions per leaf increases
considerably (manuscript in preparation). Krassilov (67) cite for the Lower Cretaceous
of Israel, 250 traces of oviposition of Zygoptera in a Dicotyledonous leaf, and from
that moment on, it is not strange to find high amounts of ovipositions, like those found
in our study.

*Geometric morphometric*

While numerous papers have contributed trace data, most descriptions of traces of
fossil eggs have been qualitative, based on linear dimensions (length-width). For the
first time, a large number of Odonata's endophytic egg traces are analysed with
geometric morphometric, except for the Romero-Lebrón (26) observation, but their
analysis was done on a single leaf. Using geometric morphometric we observed a

great diversity of oviposition trace morphotypes in the ichnospecie *P. arcuatus*. These
morphotypes are shared with *P. rectus* and *P. bifurcatus*. The diversity of
morphotypes of *P. arcuatus* could be due to the great abundance of identified
materials of this ichnotaxon, since of *P. rectus* and *P. bifurcatus* there is only one
representative of each. The greatest variation in the shape of the oviposition traces
corresponded to the shape close to the apex, and to a lesser extent to the curvature
and convexity of the trace. These three variations are also observed in the
morphology of **current** Zygoptera eggs (manuscript in preparation).

On the other hand, no particular morphotype was observed for each oviposition
substrate. Regardless of the substrate, the average shape of the traces does not
vary. Therefore, we **interpreted** that the substrate would not be exerting shape
modification pressures on the oviposition traces of the studied materials. In addition, a
marked overlap was observed in the shape of the traces regardless of their locality of
origin (LH - Lower Eocene - and RP - Middle Eocene -), so it could be inferred that
the shape of the individual traces would not depend on provenance or age. Therefore,
the variations in the shape of the individual traces would not be related to the
ichnotaxonomy, substrate used or locality of provenance or age.

In summary, the shapes of Odonata's individual egg traces are observed to be
stable over 4 million years (Lower Eocene-Middle Eocene), while individual size
varies. Insect eggs exhibit a great diversity of sizes, **these** variations are observed
within populations (17, 18) and there are even variations in the eggs that the same
female oviposes [e.g. (19)]. Therefore egg size results in a highly variable parameter
that is also observed in the traces of **Patagonia** Odonata (Lower-Middle Eocene). On
the other hand, in this work no consistent changes in egg shape are observed despite
the different substrates used for oviposition, the 4 million years considered in this
study and this is not reflected by the icnotaxonomy either. According to Legay (15)
insects have existed for hundreds of millions of years, but despite their great
diversification, the shape of their eggs shows great stability. As far as we know, this is
the first work that analyzes the change of shape of Odonata's endophytic eggs in a
scale of 4 million years and therefore, it would be the first work that empirically

reflects the stability of shape proposed by Legay (15). This could reflect that the shape of Odonata eggs, unlike their size, could have a strong evolutionary constraint already observed since the Patagonian Eocene.

Data accessibility. Our data are deposited at Dryad: <https://doi.org/10.5061/dryad.d51c5b015>. Dryad review URL <https://datadryad.org/stash/share/VFMIABxwz5HFORolcfipntAxkOMQtX0dUuC5319sJJ4>.

Acknowledgements. Thanks are due to Rubén Cúneo and Eduardo “Dudu” Ruigómez (Egidio Feruglio Museum, Trelew, Argentina) for their help and providing lab infrastructure during the visit to the MEF. We thank Claudia Tambussi, Silvia Gnaedinger, Alberto Rodrigues-Capítulo and Peter Wilf for improving the quality of this article. Funding support for the lab studies and fieldtrip came from grants: PIP 0834 from the National Research Council of Argentina (CONICET); SECyT N875 from the National University of La Plata (UNLP), PICT-2016-4297 from the National Agency of Scientific and Technological Promotion of Argentina (ANPCyT); and DEB-1556666 from the National Science Foundation of USA (NSF).

REFERENCES

1. Peñalver E, Delclòs X. Insectos del Mioceno inferior de Ribesalbes (Castellón, España). *Interacciones planta-insecto*. Treballs del Museu de Geologia de Barcelona. 2004;12:69-95.
2. Vasilenko DV. Damages on Mesozoic plants from the Transbaikalian locality Chernovskie Kopi. *Paleontological Journal*. 2005;39(6):628–33.
3. Sarzetti LC, Labandeira CC, Muzón J, Wilf P, Cúneo NR, Johnson KR, et al. Odonatan endophytic oviposition from the Eocene of Patagonia: The ichnogenus *Paleoovoidus* and implications for behavioral stasis. *Journal of Paleontology*. 2009;83(3):431-47.
4. Petrulevicius JF, Wappler T, Nel A, Rust J. The diversity of Odonata and their endophytic ovipositions from the Upper Oligocene Fossilagerstätte of Rott (Rhineland, Germany). *ZooKeys*. 2011(130):67-89.

- 5. Moisan P, Labandeira CC, Matushkina NA, Wappler T, Voigt S, Kerp H.
Lycopsid–arthropod associations and odonopteran oviposition on Triassic
herbaceous Isoetites. *Palaeogeography, Palaeoclimatology, Palaeoecology*.
2012;344-345:6-15.
- 6. Béthoux O, Galtier J, Nel A. Earliest evidence of insect endophytic oviposition.
*Palaios*. 2004 19(4):408-13.
- 7. Laaß M, Hoff C. The earliest evidence of damselfly-like endophytic oviposition
in the fossil record. *Lethaia*. 2015;48(1):115-24.
- 8. Labandeira CC, Johnson KR, Lang P. Preliminary assessment of insect
herbivory across the Cretaceous-Tertiary boundary. In: Hartman JH, Johnson KR,
Nichols DJ, editors. *The Hell Creek Formation of the northern Great Plains: Boulder*.
361. Colorado: Geological society of America special paper; 2002. p. 297–327.
- 9. Zherikhin VV. Insect trace fossils. In: Rasnitsyn AP, Quicke DLJ, editors.
*History of insects: Kluwer Academic Publishers*; 2002. p. 303-24.
- 10. Gnaedinger SC, Adami-Rodrigues K, Gallego OF. Endophytic oviposition on
leaves from the Late Triassic of northern Chile: Ichnotaxonomic, palaeobiogeographic
and palaeoenvironment considerations. *Geobios*. 2014;47(4):221-36.
- 11. Petrulevicius JF, Gutiérrez PR. New basal Odonoptera (Insecta) from the
lower Carboniferous (Serpukhovian) of Argentina. *Arquivos Entomológicos*.
2016;16:341-58.
- 12. Abbott JC. Odonata (Dragonflies and Damselflies). *Encyclopedia of Inland*
*Waters*. Austin, TX, USA: Elsevier; 2009 p. 394-404.
- 13. Corbet PS. *Dragonflies: behaviour and ecology of Odonata*. Colchester, UK:
Harley Books; 1999.
- 14. Hinton HE. *Biology of Insect Eggs*. Oxford: Pergamon Press; 1981
- 15. Legay JM. Allometry and systematics Insect egg form. *Journal of Natural*
*History*. 1977;11(5):493-9.
- 16. Church SH, Donoughe S, de Medeiros BAS, Extavour CG. Insect egg size and
shape evolve with ecology but not developmental rate. *Nature*. 2019;571(7763):58-
62.
- 17. Stearns SC. *The evolution of life histories*. Oxford, UK: Oxford University
Press; 1992.
- 18. Johnston TA, Leggett WC. Maternal and environmental gradients in the egg
size of an iteroparous fish. *Ecology*. 2002;83(7):1777–91.
- 19. Schenk K, Sondgerath D. Influence of egg size differences within egg clutches
on larval parameters in nine libellulid species (Odonata). *Ecological Entomology*.
2005;30(4):456-63.
- 20. Capinera JL. Qualitative variation in plants and insects: effect of propagule size
on ecological plasticity. *The American naturalist*. 1979 114(3):350-61.
- 21. Corkum LD, Ciborowski JJ, Poulin RG. Effects of emergence date and
maternal size on egg development and sizes of eggs and first-instar nymphs of a
semelparous aquatic insect. *Oecologia*. 1997;111(1):69-75.
- 22. Johnson DM. Behavioral ecology of larval dragonflies and damselflies. *Trends*
*in Ecology and Evolution*. 1991;6(1):8-13.
- 23. Anholt BR. Cannibalism and early instar survival in a larval damselfly.
*Oecologia*. 1994 99:60-5.

24. Hopper KR, Crowley PH, Kielman D. Density Dependence, Hatching
Synchrony, and within-Cohort Cannibalism in Young Dragonfly Larvae. *Ecology*.
1996;77(1):191-200.
25. Padeffke T, Suhling F. Temporal priority and intra-guild predation in temporary
waters: an experimental study using Namibian desert dragonflies. *Ecological*
*Entomology*. 2003 28:340–7.
26. Romero-Lebrón E, Gleiser RM, Petrulevičius JF. Geometric morphometrics to
interpret the endophytic egg-laying behavior of Odonata (Insecta) from the Eocene of
Patagonia, Argentina. *Journal of Paleontology*. 2019;93(06):1126-36.
27. González CC. Revisión taxonómica y biogeográfica de las familias de
angiospermas dominantes de la “Flora del Hunco” (Eoceno temprano), Chubut,
Argentina: Universidad de Buenos Aires, Buenos Aires, Argentina; 2009.
28. Wilf P. Rainforest conifers of Eocene Patagonia: attached cones and foliage of
the extant Southeast Asian and Australasian genus *Dacrycarpus* (Podocarpaceae).
*American journal of botany*. 2012;99(3):562-84.
29. Petrulevičius JF. Palaeoenvironmental and palaeoecological implications from
body fossils and ovipositions of Odonata from the Eocene of Patagonia, Argentina.
*Terrestrial Arthropod Reviews*. 2013;6(1-2):53-60.
30. Petrulevičius JF. First *Frenguelliidae* (Odonata) from the middle Eocene of Río
Pichileufú, Patagonia, Argentina. *Arquivos Entomológicos*. 2017;18:367-74.
31. Petrulevičius J. A new burmagomphid dragonfly from the Eocene of Patagonia,
Argentina. *Acta Palaeontologica Polonica*. 2017;62.
32. Petrulevičius JF. A New Malachite Damselfly (*Synlestidae*: Odonata) from the
Eocene of Patagonia, Argentina. *Life: The Excitement of Biology*. 2018;6(2):36-43.
33. Petrulevičius JF, Nel A. *Frenguelliidae*, a new family of dragonflies from the
earliest Eocene of Argentina (Insecta: Odonata): phylogenetic relationships within
Odonata. *Journal of Natural History*. 2003;37(24):2909-17.
34. Petrulevičius JF, Nel A. *Austroperilestidae*, a new family of damselflies from
the earliest Eocene of Argentina (Insecta: Odonata). Phylogenetic relationships within
odonata. *Journal of Paleontology*. 2005;79(4):658–62.
35. Petrulevičius JF, Nel A. Enigmatic and little known Odonata (Insecta) from the
Paleogene of Patagonia and Northwest Argentina. *Annales de la Société*
*entomologique de France (NS)*. 2013;43(3):341-7.
36. Petrulevičius JF, Nel A. A new *Frenguelliidae* (Insecta: Odonata) from the early
Eocene of Laguna del Hunco, Patagonia, Argentina. *Zootaxa*. 2013;3616:597-600.
37. Petrulevičius JF, Voisin JF. Discovery of a new genus and species of darnier
dragonfly (*Aeshnidae*: Odonata) from the lower Eocene of Laguna del Hunco,
Patagonia, Argentina. In: Nel A, Petrulevičius JF, Azar D, editors. *Fossil insects*. 46.
Paris: Special issue *Annales de la Société Entomologique de France*; 2010. p. 271-5.
38. Wilf P, Cúneo NR, Johnson KR, Hicks JF, Wing SL, Obradovich JD. High Plant
Diversity in Eocene South America: Evidence from Patagonia. 2003.
39. Wilf P, Johnson KR, Cúneo NR, Smith ME, Singer BS, Gandolfo MA. Eocene
Plant Diversity at Laguna del Hunco and Río Pichileufú, Patagonia, Argentina. 2005.
40. Wilf P, Nixon KC, Gandolfo MA, Cuneo NR. Eocene Fagaceae from Patagonia
and Gondwanan legacy in Asian rainforests. *Science*. 2019;364(6444).
41. Adams DC, Rohlf FJ, Slice DE. Geometric morphometrics: Ten years of
progress following the ‘revolution’. *Italian Journal of Zoology*. 2004;71(1):5-16.

42. Rohlf FJ. Fitting curves to outlines. In: Rohlf FJ, Bookstein FL, editors.
Proceedings of the Michigan morphometrics workshop. 2. Michigan University of the
Michigan Museum of Zoology; 1990. p. 167–77.
43. Crampton JS. Elliptic Fourier shape analysis of fossil bivalves: some practical
considerations. *Lethaia*. 1995;28 (2):179-86.
44. Swiderski DL, Zelditch ML, Fink WL. Comparability, morphometrics y
phylogenetic systematics. In: MacLeod N, Forey F, editors. *Morphology, shape y*
*phylogeny*2002. p. 67–99.
45. Iwata H, Ukai Y. SHAPE: A Computer Program Package for Quantitative
Evaluation of Biological Shapes Based on Elliptic Fourier Descriptors. *The Journal of*
*Heredity*. 2002 93(5):384-5.
46. Hammer Ø, Harper DAT. Morphometrics. In: Hammer Ø, Harper DAT, editors.
*Paleontological Data Analysis*. Oxford: Blackwell Publishing; 2006. p. 78–156.
47. Hammer Ø, Harper DAT, Ryan PD. Past: Paleontological Statistics Software
Package for Education and Data Analysis. *Palaeontologia Electronica*. 2001;4(1):9.
48. Krassilov V, Silantieva N, Lewy Z. Traumas on Fossil Leaves from the
Cretaceous of Israel Krassilov V, Rasnitsyn A, editors. Sofia-Moscow: Pensoft; 2008.
49. Waage JK. Choice and utilization of oviposition sites by female *Calopteryx*
*maculata* (Odonata: Calopterygidae). *Behavioral Ecology and Sociobiology*.
1987;20(6):439-46.
50. Buskirk RE, Sherman KJ. The influence of larval ecology on oviposition and
mating strategies in dragonflies. *Florida Entomologist*. 1985;68(1):39-51.
51. Guillermo-Ferreira R, Del-Claro K. Oviposition site selection in *Oxyagrion*
*microstigma*Selys, 1876 (Odonata: Coenagrionidae) is related to aquatic vegetation
structure. *International Journal of Odonatology*. 2011;14(3):275-9.
52. Hellmund M, Hellmund W. Eiablageverhalten fossiler Kleinlibellen (Odonata,
Zygoptera) aus dem Oberoligozän von Rott im Siebengebirge. *Stuttgarter Beiträge*
*zur Naturkunde Serie B (Geologie und Paläontologie)*. 1991;177(17):1-17.
53. Hellmund M, Hellmund W. Zum Fortpflanzungsmodus fossiler Kleinlibellen
(Insecta, Odonata, Zygoptera). *Paläontologische Zeitschrift*. 1996;70(1-2):153-70.
54. Schachat SR, Labandeira CC, Gordon J, Chaney D, Levi S, Halthore MN, et al.
Plant-Insect Interactions from Early Permian (Kungurian) Colwell Creek Pond, North-
Central Texas: The Early Spread of Herbivory in Riparian Environments. *International*
*Journal of Plant Sciences*. 2014;175(8):855-90.
55. Srivastava AK. Lower Barakar flora of Raniganj Coalfield and insect/plant
relationship india. *The Palaeobotanist*. 1987 36:138-42.
56. Hellmund M, Hellmund W. Neufunde und Ergänzungen zur
Fortpflanzungsbiologie fossiler Kleinlibellen (Insecta, Odonata, Zygoptera). *Stuttgarter*
*Beiträge zur Naturkunde Serie B (Geologie und Paläontologie)*. 2002;319(26):1-26.
57. Prevec R, Labandeira CC, Neveling J, Gastaldo RA, Looy CV, Bamford M.
Portrait of a Gondwanan ecosystem: A new late Permian fossil locality from KwaZulu-
Natal, South Africa. *Review of Palaeobotany and Palynology*. 2009;156(3-4):454-93.
58. Vassilenko DV. Traces of Interactions between Arthropods and Plants from the
Upper Permian Deposits of European Russia. In: Krassilov VA, Stage S, editors.
*Fossil insects of the Middle and Upper Permian*. 47: *Paleontological Journal*; 2013. p.
675-8.

59. Gallego J, Cúneo R, Escapa I. Plant–arthropod interactions in gymnosperm
leaves from the Early Permian of Patagonia, Argentina. *Geobios*. 2014;47(3):101-10.
60. Schachat SR, Labandeira CC, Chaney DS. Insect herbivory from early
Permian Mitchell Creek Flats of north-central Texas: Opportunism in a balanced
component community. *Palaeogeography, Palaeoclimatology, Palaeoecology*.
2015;440:830-47.
61. Grauvogel-Stamm L, Kelber KA. Plant-insect interactions and coevolution
during the Triassic in western Europe. *Paleontologia Lombarda*. 1996; 5: 5-23.
62. van Konijnenburg-van Cittert JH, Schmeißner S. Fossil insect eggs on Lower
Jurassic plant remains from Bavaria (Germany). *Palaeogeography,*
*Palaeoclimatology, Palaeoecology* 1999;152(3-4):215-23.
63. Banerji J. Evidence of Insect-plant Interactions From the Upper Gondwana
Sequence (Lower Cretaceous) in the Rajmahal Basin, India. *Gondwana Research*.
2004;7(1):205-10.
64. Donovan MP, Iglesias A, Wilf P, Labandeira CC, Cúneo NR. Diverse Plant-
Insect Associations from the Latest Cretaceous and Early Paleocene of Patagonia,
Argentina. *Ameghiniana*. 2018;55(3):303.
65. Hellmund M, Hellmund W. Fossile Zeugnisse zum Verhalten von Kleinlibellen
aus Rott. Koenigswald Wv, editor. Bonn: Rheinlandia Verlag; 1996. 57-60 p.
66. Hellmund M, Hellmund W. Eilogen von Zygopteren (Insecta, Odonata,
Coenagrionidae) in unteroligozänen Maarsedimenten von Hammerunterwiesenthal
(Freistaat Sachsen). *Abhandlungen des Staatlichen Museums für Mineralogie und*
*Geologie zu Dresden*. 1998;43(44):281-92.
67. Krassilov V, Silantieva N, Hellmund M, Hellmund W. Insect egg sets on
angiosperm leaves from the Lower Cretaceous of Negev, Israel. *Cretaceous*
*Research*. 2007;28(5):803-11.

**Supporting information labels:**

Appendix S1. Material MPEF-IC-1382. A) Original material; B) Partial photographs taken with
the magnifying glass completing the whole sheet. It has an overlapping layer in which the
contours of each trace were marked and identified with a number; C) Detail of the layer with
marked and identified traces; D) Principal component analysis graph PC 1 (59.41 %) - PC 2
(32.41 %) with 95% confidence ellipse. Scales: A) and B) 1 cm, C) 1x1 mm black square.

Appendix S2. Material MEF-IC-1376. A) Original material; B) Partial photographs taken with a
magnifying glass completing the whole leaf. It has an overlapping layer in which the contours
of each trace were marked and identified with a number; C) Detail of the layer with marked

and identified traces; D) Principal component analysis graph PC 1 (60.06 %) - PC 2 (30.30
552 %) with 95% confidence ellipse. Scales: A) and B) 1 cm, C) 1x1 mm black square.

Appendix S3. Material MEF-IC-1385. A) Original material; B) Partial photographs taken with a
magnifying glass completing the whole leaf. It has an overlapping layer in which the contours
of each trace were marked, which were identified with a number; C) Detail of the layer with
marked and identified traces; D) Principal component analysis graph PC 1 (87.44 %) - PC 2
(7.55 %) with 95% confidence ellipse. Scales: A) and B) 1 cm, C) 1x1 mm black square.

Appendix S4. Material MEFP-IC-1388. A) Original material; B) Partial photographs taken with
a magnifying glass completing the whole leaf. It has an overlapping layer in which the
contours of each trace were marked, which were identified with a number; C) Detail of the
layer with marked and identified traces; D) Principal component analysis graph PC 1 (57.06
564 %) - PC 2 (23.80 %) with 95% confidence ellipse. Outliers traces numbers: 31, 33, 40, 95, 98
and 240. Scales: 1) and 2) 1 cm, 3) 1x1 mm black square.

Appendix S5. Material MEF-IC-1378. A) Original material; B) Partial photographs taken with a
magnifying glass completing the whole leaf. It has an overlapping layer in which the contours
of each trace were marked, which were identified with a number; C) Detail of the layer with
marked and identified traces; D) Principal component analysis graph PC 1 (55.01 %) - CP 2
(27.10 %) with 95% confidence ellipse Outliers trace numbers: 45, 48, 50, 54, 71 and 72.
Scales: A) and B) 1 cm, C) 1x1 mm black square.

Figure 1. A) Image of the Dicotyledonous leaf with traces of *Odonata* endophytic eggs located along the entire leaf surface (MPEF-IC-1388), scale: 1 cm. B) Superposition of layers to the main photograph of the leaf in which the contours of each trace were marked in detail. C) Detail of an egg trace with the scale (square 1 x 1 mm) in the lower margin. D) Map of Argentina showing the localities of Río Pichileufú and Laguna del Hunco in the Patagonia.

165x160mm (300 x 300 DPI)

Figure 2. Classical morphometrics of the traces of Odonata endophytic oviposition in relation with their ichnotaxonomy (A-E) and for *Paleoovoidus arcuatus*, in relation with the age/locality (F-J). Mean values + standard errors are displayed. Different letters between columns indicate statistically significant differences ($p < 0.05$).

165x200mm (300 x 300 DPI)

Figure 3. Classical morphometrics of the oviposition traces of Odonata in relation with the substrate used. Average + standard error shown. Different letters between columns indicate statistically significant differences (p < 0.05).

165x282mm (300 x 300 DPI)

Figure 4. Variation of shape in the traces of oviposition in fossil leaves from Laguna del Hunco and Río Pichileufú according to ichnotaxon. MPEF-IC 1376 *Paleoovoidus rectus* (Laguna del Hunco); MPEF-IC 1386 *Paleoovoidus bifurcatus* (Laguna del Hunco); MPEF-IC 1388 *Paleoovoidus arcuatus* (Río Pichileufú); MPEF-IC 1378 *Paleoovoidus arcuatus* (Laguna del Hunco). The variation in shape is represented in standard deviation units (-2 and +2) and the mean shape is the mean of the Fourier coefficients for all forms analysed.

165x94mm (300 x 300 DPI)

Figure 5. Analysis of the principal components of Fourier coefficients, of the individual traces analysed in 25 samples in relation to the 3 identified ichnotaxa (black dots correspond to MPEF-IC-1382 material assigned to *P. arcuatus*).

165x88mm (300 x 300 DPI)

Figure 6. Analysis of the principal components of Fourier coefficients, of the individual traces analysed in relation to the 12 oviposition substrates identified at family or species level.

165x88mm (300 x 300 DPI)

Figure 7. Analysis of the principal components of the Fourier coefficients, of the individual traces analysed in 25 samples in relation to their origin: Laguna del Hunco (Lower Eocene) and Río Pichileufú (Middle Eocene).

165x88mm (300 x 300 DPI)

Table 1. Materials analysed: identification, locality (RP = Río Pichileufú; LH = Laguna

MPEF-IC	Location	Substrate	Ichnospecies
1386	RP	Dicotyledonous	P. arcuatus
1388	RP	Dicotyledonous	P. arcuatus
1390	RP	Dicotyledonous	P. arcuatus
1391	RP	Dicotyledonous	P. arcuatus
1393	RP	Dicotyledonous	P. arcuatus
1382	LH	Dicotyledonous	P. arcuatus
LH	Dicotyledonous	P. arcuatus
LH	Dicotyledonous	P. arcuatus
LH	Dicotyledonous	P. arcuatus
LH	Dicotyledonous	P. arcuatus
LH	Dicotyledonous	P. arcuatus
LH	Dicotyledonous	P. arcuatus
LH	Dicotyledonous	P. arcuatus
1381	LH	Malvaceae	P. arcuatus
1370	LH	Celtidaceae	Celtis ameghenoi
1374	LH	Cunoniaceae	Cupania latifolioides
1377	LH	Fabaceae	Cassia argentinensis
1378	LH		Lomatia occidentalis
1389	LH	Proteaceae	Lomatia occidentalis
LH	Sapindaceae	Schmidelia proedulis
1368	LH		Myrcia deltodea
1369	LH		Eucalyptus chubutensis
1373	LH	Myrtaceae	Eucalyptus chubutensis
1376	LH		Eucalyptus chubutensis
1392	LH		Eucalyptus chubutensis

del Hunco), paleobotanical classification, ichnotaxonomic classification, number of traces

n traces	Length (L)	Width (W)	L/W Index	Area (mm ²)	Perimeter
8	1.28 ± 0.04	0.77 ± 0.02	1.67 ± 0.05	0.75 ± 0.03	3.47 ± 0.08
240	0.87 ± 0.01	0.33 ± 0.01	2.75 ± 0.03	0.21 ± 0.01	2.02 ± 0.04
126	1.21 ± 0.01	0.46 ± 0.01	2.65 ± 0.03	0.48 ± 0.01	3.09 ± 0.03
58	0.81 ± 0.02	0.29 ± 0.01	2.89 ± 0.07	0.17 ± 0.01	1.90 ± 0.05
44	1.34 ± 0.02	0.55 ± 0.01	2.45 ± 0.05	0.54 ± 0.02	3.32 ± 0.05
18	0.99 ± 0.04	0.41 ± 0.02	2.44 ± 0.04	0.32 ± 0.03	2.45 ± 0.11
294	0.95 ± 0.01	0.35 ± 0.35	2.82 ± 0.03	0.26 ± 0.01	2.31 ± 0.03
30	0.91 ± 0.02	0.31 ± 0.01	2.95 ± 0.08	0.22 ± 0.01	2.19 ± 0.04
21	1.18 ± 0.02	0.52 ± 0.01	2.29 ± 0.05	0.50 ± 0.02	3.05 ± 0.05
36	1.12 ± 0.03	0.38 ± 0.02	3.00 ± 0.10	0.33 ± 0.02	2.68 ± 0.08
31	0.91 ± 0.03	0.33 ± 0.01	2.84 ± 0.09	0.19 ± 0.01	2.09 ± 0.08
39	1.16 ± 0.04	0.40 ± 0.02	2.97 ± 0.09	0.37 ± 0.03	2.78 ± 0.10
21	1.07 ± 0.05	0.47 ± 0.03	2.36 ± 0.09	0.39 ± 0.04	2.66 ± 0.14
5	0.79 ± 0.10	0.25 ± 0.03	3.12 ± 0.14	0.15 ± 0.03	1.83 ± 0.23
65	1.35 ± 0.02	0.47 ± 3E ⁻³	2.85 ± 0.04	0.49 ± 0.01	3.26 ± 0.03
38	1.00 ± 0.02	0.34 ± 0.01	3.06 ± 0.10	0.26 ± 0.01	2.36 ± 0.04
30	0.98 ± 0.03	0.35 ± 0.01	2.79 ± 0.08	0.25 ± 0.01	2.33 ± 0.06
84	1.10 ± 0.01	0.42 ± 4E ⁻³	2.63 ± 0.03	0.36 ± 0.01	2.71 ± 0.03
49	0.77 ± 0.02	0.32 ± 0.01	2.44 ± 0.06	0.17 ± 0.01	1.84 ± 0.05
6	1.75 ± 0.13	0.62 ± 0.03	2.86 ± 0.28	0.86 ± 0.08	4.27 ± 0.26
46	0.84 ± 0.02	0.41 ± 0.01	2.11 ± 0.06	0.24 ± 0.01	2.07 ± 0.06
4	0.79 ± 0.08	0.33 ± 0.02	2.49 ± 0.37	0.19 ± 0.02	1.94 ± 0.16
28	0.84 ± 0.04	0.35 ± 0.02	2.51 ± 0.10	0.22 ± 0.02	2.05 ± 0.10
9	0.80 ± 0.05	0.31 ± 0.02	2.63 ± 0.19	0.15 ± 0.02	1.75 ± 0.11
16	0.84 ± 0.05	0.23 ± 0.01	3.75 ± 0.20	0.12 ± 0.01	1.80 ± 0.10

and measurements (Average \pm Standard Error). Values are expressed in mm. $P. = P$

*'aleoovoidus*

Table 2. Summary table of values of the Principal Components (PC) of the Fourier co

	n	Mean	S.E.	Min	Max	Median
PC 1	25	62.87	2.39	43.33	98.81	60.06
PC 2	25	22.71	1.62	0.73	36.32	23.8
PC 3	25	7.3	0.69	0.46	14.86	6.89

1
2)efficients.

Appendix B**ROYAL SOCIETY
OPEN SCIENCE****GEOMETRIC MORPHOMETRICS OF ENDOPHYTIC
OVIPOSITION TRACES OF ODONATA (EOCENE, ARGENTINA)**

Journal:	Royal Society Open Science
Manuscript ID	RSOS-201126
Article Type:	Research
Date Submitted by the Author:	21-Aug-2020
Complete List of Authors:	Romero-Lebrón, Eugenia; IMBIV GLEISER, RAQUEL M.; IMBIV PETRULEVIČIUS, JULIÁN F.; Facultad de Ciencias Naturales; CONICET
Subject:	palaeontology < BIOLOGY
Keywords:	ENDOPHYTIC, OVIPOSITION, TRACES, ODONATA, GEOMETRIC MORPHOMETRICS
Subject Category:	Organismal and Evolutionary Biology

Author-supplied statements

Relevant information will appear here if provided.

Ethics

Does your article include research that required ethical approval or permits?:

This article does not present research with ethical considerations

Statement (if applicable):

CUST_IF_YES_ETHICS :No data available.

Data

It is a condition of publication that data, code and materials supporting your paper are made publicly available. Does your paper present new data?:

Yes

Statement (if applicable):

Our data are deposited at Dryad: <https://doi.org/10.5061/dryad.d51c5b015>.

Dryad review URL

<https://datadryad.org/stash/share/VFMIABxwz5HFORoIcfipntAxBkOMQtX0dUuUC5319sJJ4>.

Conflict of interest

I/We declare we have no competing interests

Statement (if applicable):

CUST_STATE_CONFLICT :No data available.

Authors' contributions

This paper has multiple authors and our individual contributions were as below

Statement (if applicable):

[revised manuscript text omitted]

In all the models the fulfillment of the assumptions of normality and
homocedasticity was verified, and in its defect the correction was carried out by
means of the function varident. In case of significant effects, in order to determine
which means differed, the LSD Fisher *a posteriori* test was carried out (Alfa = 0.05).

*Geometric morphometrics*

For geometric morphometric Elliptic analysis of Fourier was performed,
following the methodology described in Romero-Lebrón (26). We worked with
individual images of the contour of each of the traces (Fig. 1 C). The free statistical
package SHAPE 1.3 (45) was used to calculate Fourier coefficients. Twenty
harmonics were taken (46) (tested with a lower number of harmonics; 20 harmonics
were better adjusted to the reference shape). With the numerous variables produced,
a principal component analysis (PCA) was performed using covariance matrices. For

this, the free distribution statistical package PAST 3.15 was used (47). Finally,
PrinPrint (SHAPE 1.3 subprogram) was used to visualize the variation in shape
represented by each principal component.

11 146 **RESULTS**

A total of 1346 traces of Odonata oviposition from 24 materials were studied (traces
that were not well defined and had incomplete contours were excluded). As MPEF-
Pb-2216 has two ichnospecies in the same leaf (MPEF-IC-1376 and MPEF-IC-1392)
and the analyses were carried out based on ichnological classifications, the final
number of samples is 25.

The sample MEF-IC-1382, which was originally unidentified, was classified b
153 us as *P. arcuatus* according to Krassilov (48) because the traces follow a curved
pattern (Appendix S1). The substrate was assigned to a dicotyledonous leaf.

Based on the botanical taxonomy of the material (information provided by the
MEF except for the sample MEF-IC-1382), a great diversity of substrates used for
oviposition is observed. Traces of oviposition were found in 12 Dicotyledons without
minor identification, and in 7 families: Malvaceae, Myrtaceae, Celtidaceae,
Cunoniaceae, Fabaceae, Proteaceae, and Sapindaceae (Table 1).

Regarding the abundance of substrates used for endophytic oviposition, 50 %
of the material was only identified at the Dicotyledonous Class level; for the
Celtidaceae (*Celtis ameghenoi* Berry), Cunoniaceae (*Cupania latifolioides* Berry),
Fabaceae (*Cassia argentinensis* Berry), Malvaceae and Sapindaceae families
(*Schmidelia proedulis* Berry) 1 material of each was found. The Myrtaceae family was
the most represented (*Myrcia deltodea* Berry, *Eucalyptus chubutensis* Berry) followed
by Proteaceae (*Lomatia occidentalis* Berry) (Table 1).

As for the ichnotaxonomic classification already attributed to the reviewed
materials, there were three ichnospecies: *Paleoovoidus arcuatus*, *Paleoovoidus*
*bifurcatus* and *Paleoovoidus rectus* (Table 1). The ichnospecies *P. arcuatus* was the

most frequent ($n = 22$) representing 91.67 % of total materials, followed by *P.*
*bifurcatus* and *P. rectus*, each with only 1 specimen, and one unidentified material
now assigned to *P. arcuatus*. *Paleoovoidus arcuatus* is present in a great diversity of
substrates. On one occasion, *P. arcuatus* coexists with *P. rectus* in a single specimen
of the family Myrtaceae (*Eucalyptus chubutensis*) [Romero-Lebrón (26) analyse this
material in detail]. The only specimen of *P. bifurcatus* is found in the only sample with
traces of the family Sapindaceae (*Schmidelia proedulis*) (Table 1).

*Classical Morphometrics*

*Relationship of metrics with provenance (age), ichnotaxonomic classification and*
*oviposition substrate*. The traces from Río Pichileufú (Middle Eocene) were
significantly wider and of larger area than those from Laguna del Hunco (Lower
Eocene) ($p < 0.05$). The traces of greater length, area and perimeter are those of *P.*
*bifurcatus* (MPEF-IC-1385: 1.75 ± 0.13 mm), while those of *P. arcuatus* and *P. rectus*
do not differ from each other ($p < 0.05$). Material MPEF-IC-1382 (which we assigned
to *P. arcuatus*) has a trace length similar to *P. arcuatus* and *P. rectus* (Fig. 2). The
substrate with traces of greater ($p < 0.05$) length, area and perimeter is *Schmidelia*
*proedulis* (classified as *P. bifurcatus*); while the other substrates in general show
relatively little variation in all dimensions (Fig. 3). There were also no significant
differences in trace dimensions between the families Myrtaceae and Proteaceae.

*Geometric Morphometrics*

For each of the 5 samples a principal component analysis (PCA) of their Fourier
coefficients was performed to describe the shape of the individual traces (Fig. Appendix S2-S5).
The principal component 1 explained between 43.33 % and 98.81 % of the total variation of the data
(Table 2). In 84 % of the material the shape close
to the apex was discriminated, while in 12 % the apex shape was discriminated. The

principal component 2 explained between 0.73 % and 36.32 % of the total variation of
the data, discriminating the curvature of the trace.

*Ichnotaxonomy and geometric morphometrics*

The PCA of Fourier coefficients in relation to trace ichnotaxonomy reduced their
variability to three principal components (PC) which together explain 92.61 % of the
total variation observed in the shape. PC 1, PC 2 and PC 3 respectively explain 58.06
205 %, 26.83 % and 7.72 % of the total variation. A great diversity of oviposition trace
morphotypes is observed in *P. arcuatus*. However, no particular trace shape was
detected for each taxon, as the morphotypes observed in *P. arcuatus* are shared with
*P. rectus* and *P. bifurcatus* (Fig. 5).

*Substrate diversity and geometric morphometrics*

The first three principal components of the PCA of Fourier coefficients in relation to
substrate diversity together explain 92.66 % of the total variance in the shape. PC 1,
2 and 3 explain 64.54 %, 21.08 % and 7.04 % of the total variation, respectively. We
assessed the 12 materials identified at family or species level. Those assigned only to
the category "Dicotyledonous" were excluded in this instance of analysis (see Table 1
for details of the materials). No particular morphotype is observed for each substrate
(Fig. 6).

*Comparison between Lower and Middle Eocene*

The analysis of principal components on the Fourier coefficients considering the
ages/localities of the samples reduced the variability to three principal components
(PC) that together capture 92.61 % of the total variance observed in the shape. PC 1,
2 and 3 explain 58.06 %, 26.83 % and 7.72 % of the total variation, respectively. A
large overlap of morphotypes is observed, regardless of age/locality (Fig. 7).

1
2
3 225
45 226 **DISCUSSION**
67
*Substrate*

The choice of substrate and environment that adults choose as a place for the
development of their offspring is complex and important, as it will influence the
reproductive success of the species (49). Vegetation composition is considered to be
an important indicator for current odonates for habitat selection of future larvae (50,
51). In fossil material, there are some authors assigning substrate preferences for
oviposition to certain families of Zygoptera. According to Hellmund & Hellmund (52,
53) Lestidae females often show substrate preference for certain plants such as
*Daphnogene* leaves (Lauraceae), while Coenagrionidae females, however, do not
appear to prefer a plant type in the fossil record. In this respect, we have to be careful
to make attributions to Recent families for ovipositions in the Eocene as there are
extinct families of Odonata (29), like Frenguelliidae, Austroperilestidae and
Burmagomphidae in Laguna del Hunco and Río Pichileufú (30, 31, 33, 36) that could
be also the producers (29). Concerning a more reliable attribution of ovipositions to a
certain genus or family of plants, in our studies, it is difficult to assign a preference for
substrate because most plant taxa were represented by one or a few samples.
However, all substrates with traces of oviposition analysed correspond to
Dicotyledonea. The most represented family is Myrtaceae, being *Eucalyptus*
*chubutensis* (Myrtaceae) and *Lomatia occidentalis* (Proteaceae) the most abundant
species. Wilf (38, 39) analysed in detail the diversity of the flora of Laguna del Hunco
and Río Pichileufú based on more than 3500 fossil materials, providing a list of the
thirty most abundant plant species of Laguna del Hunco. The substrates in which
traces were detected belong to taxa of plants included in this list, therefore, the
abundance of substrates chosen for oviposition, agrees with the relative abundance
in the environment, then there would be no evidence of a pressure of selection of
substrate. On the other hand, *Myrcia deltodea* is not one of the 30 most frequent
plants, opening the question as to whether it reflects a selection by this substrate.

*Classical Morphometrics*

Historically, linear measurements have been widely used to describe fossil structures.
Sarzetti (3) provide measurements of some of the materials analysed in this study
that differ from our estimates. The differences between Sarzetti (3) and our
measurements could be due in part to the fact that they considered different numbers
of traces, for example, in the material identified as *P. rectus* (MPEF-IC-1376) they
considered twelve traces, while we identified thirteen. However, four of them were
excluded from the analyses because they were incomplete. In the case of *P.*
*arcuatus*, the differences could be due to the fact that they do not consider the
material MPEF-IC-1382, but instead material (USNM 40389) belonging to the
National Museum of Natural History, Smithsonian Institution (Washington, DC) not
evaluated by us. In the case of *P. bifurcatus*, the differences observed could be due
to different criteria when measuring the trace. In fact, differences of criteria in the
measurement are a frequent problem for those who work with linear measurements
that could be reduced using the technique of geometric morphometric.

*Ichnotaxonomy*

*Paleoovoidus arcuatus* is the ichnotaxa most abundant in Laguna del Hunco and Rio
Pichileufú. The ichnogenus *Paleoovoidus* Vasilenko (2) was originally described for
traces of oviposition belonging to the Upper Jurassic - Lower Cretaceous of Russia.
Since then it is presented in great abundance, mainly in the Paleogene and Neogene
(manuscript in preparation). Authors such as Sarzetti (3) and Petrulevičius (4, 29),
cite *P. arcuatus* in the Eocene and Oligocene, but there could be numerous
synonyms if we consider only the spatial arrangement of the set of traces (e.g.
"Coenagrionidae Type" or DT54 and DT 100) that would identify it from the Permian
[e.g. (54)] and even from the Carboniferous [e.g. (6)]. On one occasion *P. arcuatus*
and *P. rectus* were presented in the same leaf [MPEF-Pb-2216 (MPEF-IC-1376 and
MPEF-IC-1392)], being interpreted as traces made by one female (26).

Of the three ichnotaxa present, traces of *P. bifurcatus* were the longest. This
ichnospecies is mentioned for the first time in Sarzetti (3). Oviposition traces of insects
with a pattern similar to *P. bifurcatus* are found in the Lower Permian of India (55), but
the size of the traces is considerably smaller (0.1 to 0.5 mm). Later, Hellmund &
Hellmund (52, 53, 56) and Petrulevičius (4) report traces of Odonata oviposition in
Upper Oligocene materials from Germany with this type of spatial arrangement, also
in smaller size (0.5 to 0.9 mm). Considering only the spatial arrangement of the traces
of insect oviposition, in the "Curved", "Straight" and "Bifurcated" arrangement, the
"Bifurcated" is recorded from Permian to Oligocene (except Paleocene) (manuscript
in preparation). On the other hand, traces of oviposition of insects with a similar
length range are recorded in the Permian (57-60), Triassic (10, 61), Jurassic (62),
Cretaceous (63), Paleocene (64), Oligocene (53, 65, 66) and Miocene (56), but in
none of these cases do the traces have spatial arrangement similar to *P. bifurcatus*.
Therefore, the characteristics of this ichnospecies could be unique to Argentinian
Patagonia, although we would need more samples to reinforce this hypothesis.

The number of oviposition traces per leaf in the studied materials is very
variable, with a minimum - maximum range of 5 to 294 traces. In the bibliography,
from Cretaceous onwards the number of endophytic ovipositions per leaf increases
considerably (manuscript in preparation). Krassilov (67) cite for the Lower Cretaceous
of Israel, 250 traces of oviposition of Zygoptera in a Dicotyledonous leaf, and from
that moment on, it is not strange to find high amounts of ovipositions, like those found
in our study.

*Geometric morphometric*

While numerous papers have contributed trace data, most descriptions of traces of
fossil eggs have been qualitative, based on linear dimensions (length-width). For the
first time, a large number of Odonata's endophytic egg traces are analysed with
geometric morphometric, except for the Romero-Lebrón (26) observation, but their
analysis was done on a single leaf. Using geometric morphometric we observed a

great diversity of oviposition trace morphotypes in the ichnospecie *P. arcuatus*. These
morphotypes are shared with *P. rectus* and *P. bifurcatus*. The diversity of
morphotypes of *P. arcuatus* could be due to the great abundance of identified
materials of this ichnotaxon, since of *P. rectus* and *P. bifurcatus* there is only one
representative of each. The greatest variation in the shape of the oviposition traces
corresponded to the shape close to the apex, and to a lesser extent to the curvature
and convexity of the trace. These three variations are also observed in the
morphology of current Zygoptera eggs (manuscript in preparation).

On the other hand, no particular morphotype was observed for each oviposition
substrate. Regardless of the substrate, the average shape of the traces does not
vary. Therefore, we interpreted that the substrate would not be exerting shape
modification pressures on the oviposition traces of the studied materials. In addition, a
marked overlap was observed in the shape of the traces regardless of their locality of
origin (LH - Lower Eocene - and RP - Middle Eocene -), so it could be inferred that
the shape of the individual traces would not depend on provenance or age. Therefore,
the variations in the shape of the individual traces would not be related to the
ichnotaxonomy, substrate used or locality of provenance or age.

In summary, the shapes of Odonata's individual egg traces are observed to be
stable over 4 million years (Lower Eocene-Middle Eocene), while individual size
varies. Insect eggs exhibit a great diversity of sizes, these variations are observed
within populations (17, 18) and there are even variations in the eggs that the same
female oviposes [e.g. (19)]. Therefore egg size results in a highly variable parameter
that is also observed in the traces of Patagonia Odonata (Lower-Middle Eocene). On
the other hand, in this work no consistent changes in egg shape are observed despite
the different substrates used for oviposition, the 4 million years considered in this
study and this is not reflected by the icnotaxonomy either. According to Legay (15)
insects have existed for hundreds of millions of years, but despite their great
diversification, the shape of their eggs shows great stability. As far as we know, this is
the first work that analyzes the change of shape of Odonata's endophytic eggs in a
scale of 4 million years and therefore, it would be the first work that empirically

reflects the stability of shape proposed by Legay (15). This could reflect that the
shape of Odonata eggs, unlike their size, could have a strong evolutionary constrain
already observed since the Patagonian Eocene.

*Data accessibility.* Our data are deposited at Dryad:
<https://doi.org/10.5061/dryad.d51c5b015>. Dryad review URL
<https://datadryad.org/stash/share/VFMIABxwz5HFORolcfipntAxkOMQtX0dUuC5319s>
JJ4.

*Acknowledgements.* Thanks are due to Rubén Cúneo and Eduardo “Dudu” Ruigómez
(Egidio Feruglio Museum, Trelew, Argentina) for their help and providing lab
infrastructure during the visit to the MEF. We thank Claudia Tambussi, Silvia
Gnaedinger, Alberto Rodrigues-Capítulo and Peter Wilf for improving the quality of
this article. Funding support for the lab studies and fieldtrip came from grants: PIP
0834 from the National Research Council of Argentina (CONICET); SECyT N875
from the National University of La Plata (UNLP), PICT-2016-4297 from the National
Agency of Scientific and Technological Promotion of Argentina (ANPCyT); and DEB-
1556666 from the National Science Foundation of USA (NSF).

**REFERENCES**

1. Peñalver E, Delclòs X. Insectos del Mioceno inferior de Ribesalbes (Castellón,
España). *Interacciones planta-insecto*. Treballs del Museu de Geologia de Barcelona.
2004;12:69-95.
2. Vasilenko DV. Damages on Mesozoic plants from the Transbaikalian locality
Chernovskie Kopi. *Paleontological Journal*. 2005;39(6):628–33.
3. Sarzetti LC, Labandeira CC, Muzón J, Wilf P, Cúneo NR, Johnson KR, et al.
Odonatan endophytic oviposition from the Eocene of Patagonia: The ichnogenus
*Paleoovoidus* and implications for behavioral stasis. *Journal of Paleontology*.
2009;83(3):431-47.
4. Petrulevicius JF, Wappler T, Nel A, Rust J. The diversity of Odonata and their
endophytic ovipositions from the Upper Oligocene Fossilagerstätte of Rott
(Rhineland, Germany). *ZooKeys*. 2011(130):67-89.

- 5. Moisan P, Labandeira CC, Matushkina NA, Wappler T, Voigt S, Kerp H.
Lycopsid–arthropod associations and odonopteran oviposition on Triassic
herbaceous Isoetites. *Palaeogeography, Palaeoclimatology, Palaeoecology*.
2012;344-345:6-15.
- 6. Béthoux O, Galtier J, Nel A. Earliest evidence of insect endophytic oviposition.
. *Palaios*. 2004 19(4):408-13.
- 7. Laaß M, Hoff C. The earliest evidence of damselfly-like endophytic oviposition
in the fossil record. *Lethaia*. 2015;48(1):115-24.
- 8. Labandeira CC, Johnson KR, Lang P. Preliminary assessment of insect
herbivory across the Cretaceous-Tertiary boundary. In: Hartman JH, Johnson KR,
Nichols DJ, editors. *The Hell Creek Formation of the northern Great Plains: Boulder*.
361. Colorado: Geological society of America special paper; 2002. p. 297–327.
- 9. Zherikhin VV. Insect trace fossils. In: Rasnitsyn AP, Quicke DLJ, editors.
*History of insects*: Kluwer Academic Publishers; 2002. p. 303-24.
- 10. Gnaedinger SC, Adami-Rodrigues K, Gallego OF. Endophytic oviposition on
leaves from the Late Triassic of northern Chile: Ichnotaxonomic, palaeobiogeographic
and palaeoenvironment considerations. *Geobios*. 2014;47(4):221-36.
- 11. Petrulevicius JF, Gutiérrez PR. New basal Odonoptera (Insecta) from the
lower Carboniferous (Serpukhovian) of Argentina. *Arquivos Entomológicos*.
2016;16:341-58.
- 12. Abbott JC. Odonata (Dragonflies and Damselflies). *Encyclopedia of Inland*
*Waters*. Austin, TX, USA: Elsevier; 2009 p. 394-404.
- 13. Corbet PS. *Dragonflies: behaviour and ecology of Odonata*. Colchester, UK:
Harley Books; 1999.
- 14. Hinton HE. *Biology of Insect Eggs*. Oxford: Pergamon Press; 1981
- 15. Legay JM. Allometry and systematics Insect egg form. *Journal of Natural*
*History*. 1977;11(5):493-9.
- 16. Church SH, Donoughe S, de Medeiros BAS, Extavour CG. Insect egg size and
shape evolve with ecology but not developmental rate. *Nature*. 2019;571(7763):58-
62.
- 17. Stearns SC. *The evolution of life histories*. Oxford, UK: Oxford University
Press; 1992.
- 18. Johnston TA, Leggett WC. Maternal and environmental gradients in the egg
size of an iteroparous fish. *Ecology*. 2002;83(7):1777–91.
- 19. Schenk K, Sondgerath D. Influence of egg size differences within egg clutches
on larval parameters in nine libellulid species (Odonata). *Ecological Entomology*.
2005;30(4):456-63.
- 20. Capinera JL. Qualitative variation in plants and insects: effect of propagule size
on ecological plasticity. *The American naturalist*. 1979 114(3):350-61.
- 21. Corkum LD, Ciborowski JJ, Poulin RG. Effects of emergence date and
maternal size on egg development and sizes of eggs and first-instar nymphs of a
semelparous aquatic insect. *Oecologia*. 1997;111(1):69-75.
- 22. Johnson DM. Behavioral ecology of larval dragonflies and damselflies. *Trends*
*in Ecology and Evolution*. 1991;6(1):8-13.
- 23. Anholt BR. Cannibalism and early instar survival in a larval damselfly.
*Oecologia*. 1994 99:60-5.

24. Hopper KR, Crowley PH, Kielman D. Density Dependence, Hatching
Synchrony, and within-Cohort Cannibalism in Young Dragonfly Larvae. *Ecology*.
1996;77(1):191-200.
25. Padeffke T, Suhling F. Temporal priority and intra-guild predation in temporary
waters: an experimental study using Namibian desert dragonflies. *Ecological*
*Entomology*. 2003 28:340–7.
26. Romero-Lebrón E, Gleiser RM, Petrulevičius JF. Geometric morphometrics to
interpret the endophytic egg-laying behavior of Odonata (Insecta) from the Eocene of
Patagonia, Argentina. *Journal of Paleontology*. 2019;93(06):1126-36.
27. González CC. Revisión taxonómica y biogeográfica de las familias de
angiospermas dominantes de la “Flora del Hunco” (Eoceno temprano), Chubut,
Argentina: Universidad de Buenos Aires, Buenos Aires, Argentina; 2009.
28. Wilf P. Rainforest conifers of Eocene Patagonia: attached cones and foliage of
the extant Southeast Asian and Australasian genus *Dacrycarpus* (Podocarpaceae).
*American journal of botany*. 2012;99(3):562-84.
29. Petrulevičius JF. Palaeoenvironmental and palaeoecological implications from
body fossils and ovipositions of Odonata from the Eocene of Patagonia, Argentina.
*Terrestrial Arthropod Reviews*. 2013;6(1-2):53-60.
30. Petrulevičius JF. First *Frenguelliidae* (Odonata) from the middle Eocene of Río
Pichileufú, Patagonia, Argentina. *Arquivos Entomológicos*. 2017;18:367-74.
31. Petrulevičius J. A new burmagomphid dragonfly from the Eocene of Patagonia,
Argentina. *Acta Palaeontologica Polonica*. 2017;62.
32. Petrulevičius JF. A New Malachite Damselfly (*Synlestidae*: Odonata) from the
Eocene of Patagonia, Argentina. *Life: The Excitement of Biology*. 2018;6(2):36-43.
33. Petrulevičius JF, Nel A. *Frenguelliidae*, a new family of dragonflies from the
earliest Eocene of Argentina (Insecta: Odonata): phylogenetic relationships within
Odonata. *Journal of Natural History*. 2003;37(24):2909-17.
34. Petrulevičius JF, Nel A. *Austroperilestidae*, a new family of damselflies from
the earliest Eocene of Argentina (Insecta: Odonata). Phylogenetic relationships within
odonata. *Journal of Paleontology*. 2005;79(4):658–62.
35. Petrulevičius JF, Nel A. Enigmatic and little known Odonata (Insecta) from the
Paleogene of Patagonia and Northwest Argentina. *Annales de la Société*
*entomologique de France (NS)*. 2013;43(3):341-7.
36. Petrulevičius JF, Nel A. A new *Frenguelliidae* (Insecta: Odonata) from the early
Eocene of Laguna del Hunco, Patagonia, Argentina. *Zootaxa*. 2013;3616:597-600.
37. Petrulevičius JF, Voisin JF. Discovery of a new genus and species of darnier
dragonfly (*Aeshnidae*: Odonata) from the lower Eocene of Laguna del Hunco,
Patagonia, Argentina. In: Nel A, Petrulevičius JF, Azar D, editors. *Fossil insects*. 46.
Paris: Special issue *Annales de la Société Entomologique de France*; 2010. p. 271-5.
38. Wilf P, Cúneo NR, Johnson KR, Hicks JF, Wing SL, Obradovich JD. High Plant
Diversity in Eocene South America: Evidence from Patagonia. 2003.
39. Wilf P, Johnson KR, Cúneo NR, Smith ME, Singer BS, Gandolfo MA. Eocene
Plant Diversity at Laguna del Hunco and Río Pichileufú, Patagonia, Argentina. 2005.
40. Wilf P, Nixon KC, Gandolfo MA, Cuneo NR. Eocene Fagaceae from Patagonia
and Gondwanan legacy in Asian rainforests. *Science*. 2019;364(6444).
41. Adams DC, Rohlf FJ, Slice DE. Geometric morphometrics: Ten years of
progress following the ‘revolution’. *Italian Journal of Zoology*. 2004;71(1):5-16.

42. Rohlf FJ. Fitting curves to outlines. In: Rohlf FJ, Bookstein FL, editors.
Proceedings of the Michigan morphometrics workshop. 2. Michigan University of the
Michigan Museum of Zoology; 1990. p. 167–77.
43. Crampton JS. Elliptic Fourier shape analysis of fossil bivalves: some practical
considerations. *Lethaia*. 1995;28 (2):179-86.
44. Swiderski DL, Zelditch ML, Fink WL. Comparability, morphometrics y
phylogenetic systematics. In: MacLeod N, Forey F, editors. *Morphology, shape y*
*phylogeny*2002. p. 67–99.
45. Iwata H, Ukai Y. SHAPE: A Computer Program Package for Quantitative
Evaluation of Biological Shapes Based on Elliptic Fourier Descriptors. *The Journal of*
*Heredity*. 2002 93(5):384-5.
46. Hammer Ø, Harper DAT. *Morphometrics*. In: Hammer Ø, Harper DAT, editors.
*Paleontological Data Analysis*. Oxford: Blackwell Publishing; 2006. p. 78–156.
47. Hammer Ø, Harper DAT, Ryan PD. *Past: Paleontological Statistics Software*
*Package for Education and Data Analysis*. *Palaeontologia Electronica*. 2001;4(1):9.
- 48. Krassilov V, Silantieva N, Lewy Z. *Traumas on Fossil Leaves from the*
*Cretaceous of Israel* Krassilov V, Rasnitsyn A, editors. Sofia-Moscow: Pensoft; 2008.
49. Waage JK. Choice and utilization of oviposition sites by female *Calopteryx*
*maculata* (Odonata: Calopterygidae). *Behavioral Ecology and Sociobiology*.
1987;20(6):439-46.
50. Buskirk RE, Sherman KJ. The influence of larval ecology on oviposition and
mating strategies in dragonflies. *Florida Entomologist*. 1985;68(1):39-51.
51. Guillermo-Ferreira R, Del-Claro K. Oviposition site selection in *Oxyagrion*
*microstigma*Selys, 1876 (Odonata: Coenagrionidae) is related to aquatic vegetation
structure. *International Journal of Odonatology*. 2011;14(3):275-9.
52. Hellmund M, Hellmund W. Eiablageverhalten fossiler Kleinlibellen (Odonata,
Zygoptera) aus dem Oberoligozän von Rott im Siebengebirge. *Stuttgarter Beiträge*
*zur Naturkunde Serie B (Geologie und Paläontologie)*. 1991;177(17):1-17.
53. Hellmund M, Hellmund W. Zum Fortpflanzungsmodus fossiler Kleinlibellen
(Insecta, Odonata, Zygoptera). *Paläontologische Zeitschrift*. 1996;70(1-2):153-70.
54. Schachat SR, Labandeira CC, Gordon J, Chaney D, Levi S, Halthore MN, et al.
*Plant-Insect Interactions from Early Permian (Kungurian) Colwell Creek Pond, North-*
*Central Texas: The Early Spread of Herbivory in Riparian Environments*. *International*
*Journal of Plant Sciences*. 2014;175(8):855-90.
55. Srivastava AK. Lower Barakar flora of Raniganj Coalfield and insect/plant
relationship india. *The Palaeobotanist*. 1987 36:138-42.
56. Hellmund M, Hellmund W. Neufunde und Ergänzungen zur
Fortpflanzungsbiologie fossiler Kleinlibellen (Insecta, Odonata, Zygoptera). *Stuttgarter*
*Beiträge zur Naturkunde Serie B (Geologie und Paläontologie)*. 2002;319(26):1-26.
57. Prevec R, Labandeira CC, Neveling J, Gastaldo RA, Looy CV, Bamford M.
*Portrait of a Gondwanan ecosystem: A new late Permian fossil locality from KwaZulu-*
*Natal, South Africa*. *Review of Palaeobotany and Palynology*. 2009;156(3-4):454-93.
58. Vassilenko DV. *Traces of Interactions between Arthropods and Plants from the*
*Upper Permian Deposits of European Russia*. In: Krassilov VA, Stage S, editors.
*Fossil insects of the Middle and Upper Permian*. 47: *Paleontological Journal*; 2013. p.
675-8.

59. Gallego J, Cúneo R, Escapa I. Plant–arthropod interactions in gymnosperm
leaves from the Early Permian of Patagonia, Argentina. *Geobios*. 2014;47(3):101-10.
60. Schachat SR, Labandeira CC, Chaney DS. Insect herbivory from early
Permian Mitchell Creek Flats of north-central Texas: Opportunism in a balanced
component community. *Palaeogeography, Palaeoclimatology, Palaeoecology*.
2015;440:830-47.
61. Grauvogel-Stamm L, Kelber KA. Plant-insect interactions and coevolution
during the Triassic in western Europe. *Paleontologia Lombarda*. 1996; 5: 5-23.
62. van Konijnenburg-van Cittert JH, Schmeißner S. Fossil insect eggs on Lower
Jurassic plant remains from Bavaria (Germany). *Palaeogeography,*
*Palaeoclimatology, Palaeoecology* 1999;152(3-4):215-23.
63. Banerji J. Evidence of Insect-plant Interactions From the Upper Gondwana
Sequence (Lower Cretaceous) in the Rajmahal Basin, India. *Gondwana Research*.
2004;7(1):205-10.
64. Donovan MP, Iglesias A, Wilf P, Labandeira CC, Cúneo NR. Diverse Plant-
Insect Associations from the Latest Cretaceous and Early Paleocene of Patagonia,
Argentina. *Ameghiniana*. 2018;55(3):303.
65. Hellmund M, Hellmund W. Fossile Zeugnisse zum Verhalten von Kleinlibellen
aus Rott. Koenigswald Wv, editor. Bonn: Rheinlandia Verlag; 1996. 57-60 p.
66. Hellmund M, Hellmund W. Eilogen von Zygopteren (Insecta, Odonata,
Coenagrionidae) in unteroligozänen Maarsedimenten von Hammerunterwiesenthal
(Freistaat Sachsen). *Abhandlungen des Staatlichen Museums für Mineralogie und*
*Geologie zu Dresden*. 1998;43(44):281-92.
67. Krassilov V, Silantieva N, Hellmund M, Hellmund W. Insect egg sets on
angiosperm leaves from the Lower Cretaceous of Negev, Israel. *Cretaceous*
*Research*. 2007;28(5):803-11.

**Supporting information labels:**

Appendix S1. Material MPEF-IC-1382. A) Original material; B) Partial photographs taken with
the magnifying glass completing the whole sheet. It has an overlapping layer in which the
contours of each trace were marked and identified with a number; C) Detail of the layer with
marked and identified traces; D) Principal component analysis graph PC 1 (59.41 %) - PC 2
(32.41 %) with 95% confidence ellipse. Scales: A) and B) 1 cm, C) 1x1 mm black square.

Appendix S2. Material MEF-IC-1376. A) Original material; B) Partial photographs taken with a
magnifying glass completing the whole leaf. It has an overlapping layer in which the contours
of each trace were marked and identified with a number; C) Detail of the layer with marked

and identified traces; D) Principal component analysis graph PC 1 (60.06 %) - PC 2 (30.30
552 %) with 95% confidence ellipse. Scales: A) and B) 1 cm, C) 1x1 mm black square.

Appendix S3. Material MEF-IC-1385. A) Original material; B) Partial photographs taken with a
magnifying glass completing the whole leaf. It has an overlapping layer in which the contours
of each trace were marked, which were identified with a number; C) Detail of the layer with
marked and identified traces; D) Principal component analysis graph PC 1 (87.44 %) - PC 2
(7.55 %) with 95% confidence ellipse. Scales: A) and B) 1 cm, C) 1x1 mm black square.

Appendix S4. Material MEFP-IC-1388. A) Original material; B) Partial photographs taken with
a magnifying glass completing the whole leaf. It has an overlapping layer in which the
contours of each trace were marked, which were identified with a number; C) Detail of the
layer with marked and identified traces; D) Principal component analysis graph PC 1 (57.06
564 %) - PC 2 (23.80 %) with 95% confidence ellipse. Outliers traces numbers: 31, 33, 40, 95, 98
and 240. Scales: 1) and 2) 1 cm, 3) 1x1 mm black square.

Appendix S5. Material MEF-IC-1378. A) Original material; B) Partial photographs taken with a
magnifying glass completing the whole leaf. It has an overlapping layer in which the contours
of each trace were marked, which were identified with a number; C) Detail of the layer with
marked and identified traces; D) Principal component analysis graph PC 1 (55.01 %) - CP 2
(27.10 %) with 95% confidence ellipse Outliers trace numbers: 45, 48, 50, 54, 71 and 72.
Scales: A) and B) 1 cm, C) 1x1 mm black square.

Figure 1. A) Image of the Dicotyledonous leaf with traces of Odonata endophytic eggs located along the entire leaf surface (MPEF-IC-1388), scale: 1 cm. B) Superposition of layers to the main photograph of the leaf in which the contours of each trace were marked in detail. C) Detail of an egg trace with the scale (square 1 x 1 mm) in the lower margin. D) Map of Argentina showing the localities of Río Pichileufú and Laguna del Hunco in the Patagonia.

165x160mm (300 x 300 DPI)

Figure 2. Classical morphometrics of the traces of Odonata endophytic oviposition in relation with their ichnotaxonomy (A-E) and for *Paleoovoidus arcuatus*, in relation with the age/locality (F-J). Mean values + standard errors are displayed. Different letters between columns indicate statistically significant differences ($p < 0.05$).

165x200mm (300 x 300 DPI)

Figure 3. Classical morphometrics of the oviposition traces of Odonata in relation with the substrate used. Average + standard error shown. Different letters between columns indicate statistically significant differences (p < 0.05).

165x282mm (300 x 300 DPI)

Figure 4. Variation of shape in the traces of oviposition in fossil leaves from Laguna del Hunco and Río Pichileufú according to ichnotaxon. MPEF-IC 1376 *Paleoovoidus rectus* (Laguna del Hunco); MPEF-IC 1386 *Paleoovoidus bifurcatus* (Laguna del Hunco); MPEF-IC 1388 *Paleoovoidus arcuatus* (Río Pichileufú); MPEF-IC 1378 *Paleoovoidus arcuatus* (Laguna del Hunco). The variation in shape is represented in standard deviation units (-2 and +2) and the mean shape is the mean of the Fourier coefficients for all forms analysed.

165x94mm (300 x 300 DPI)

Figure 5. Analysis of the principal components of Fourier coefficients, of the individual traces analysed in 25 samples in relation to the 3 identified ichnotaxa (black dots correspond to MPEF-IC-1382 material assigned to *P. arcuatus*).

165x88mm (300 x 300 DPI)

Figure 6. Analysis of the principal components of Fourier coefficients, of the individual traces analysed in relation to the 12 oviposition substrates identified at family or species level.

165x88mm (300 x 300 DPI)

Figure 7. Analysis of the principal components of the Fourier coefficients, of the individual traces analysed in 25 samples in relation to their origin: Laguna del Hunco (Lower Eocene) and Río Pichileufú (Middle Eocene).

165x88mm (300 x 300 DPI)

Table 1. Materials analysed: identification, locality (RP = Río Pichileufú; LH = Laguna

MPEF-IC	Location	Substrate	Ichnospecies
RP	Dicotyledonous	P. arcuatus
RP	Dicotyledonous	P. arcuatus
RP	Dicotyledonous	P. arcuatus
RP	Dicotyledonous	P. arcuatus
RP	Dicotyledonous	P. arcuatus
1382	LH	Dicotyledonous	P. arcuatus
LH	Dicotyledonous	P. arcuatus
LH	Dicotyledonous	P. arcuatus
LH	Dicotyledonous	P. arcuatus
LH	Dicotyledonous	P. arcuatus
LH	Dicotyledonous	P. arcuatus
LH	Dicotyledonous	P. arcuatus
LH	Dicotyledonous	P. arcuatus
1381	LH	Malvaceae	P. arcuatus
1370	LH	Celtidaceae	Celtis ameghenoi
1374	LH	Cunoniaceae	Cupania latifolioides
1377	LH	Fabaceae	Cassia argentinensis
1378	LH		Lomatia occidentalis
1389	LH	Proteaceae	Lomatia occidentalis
1385	LH	Sapindaceous	Schmidelia proedulis
1368	LH		Myrcia deltodea
1369	LH		Eucalyptus chubutensis
1373	LH	Myrtaceae	Eucalyptus chubutensis
1376	LH		Eucalyptus chubutensis
1392	LH		Eucalyptus chubutensis

del Hunco), paleobotanical classification, ichnotaxonomic classification, number of traces

n traces	Length (L)	Width (W)	L/W Index	Area (mm ²)	Perimeter
8	1.28 ± 0.04	0.77 ± 0.02	1.67 ± 0.05	0.75 ± 0.03	3.47 ± 0.08
240	0.87 ± 0.01	0.33 ± 0.01	2.75 ± 0.03	0.21 ± 0.01	2.02 ± 0.04
126	1.21 ± 0.01	0.46 ± 0.01	2.65 ± 0.03	0.48 ± 0.01	3.09 ± 0.03
58	0.81 ± 0.02	0.29 ± 0.01	2.89 ± 0.07	0.17 ± 0.01	1.90 ± 0.05
44	1.34 ± 0.02	0.55 ± 0.01	2.45 ± 0.05	0.54 ± 0.02	3.32 ± 0.05
18	0.99 ± 0.04	0.41 ± 0.02	2.44 ± 0.04	0.32 ± 0.03	2.45 ± 0.11
294	0.95 ± 0.01	0.35 ± 0.35	2.82 ± 0.03	0.26 ± 0.01	2.31 ± 0.03
30	0.91 ± 0.02	0.31 ± 0.01	2.95 ± 0.08	0.22 ± 0.01	2.19 ± 0.04
21	1.18 ± 0.02	0.52 ± 0.01	2.29 ± 0.05	0.50 ± 0.02	3.05 ± 0.05
36	1.12 ± 0.03	0.38 ± 0.02	3.00 ± 0.10	0.33 ± 0.02	2.68 ± 0.08
31	0.91 ± 0.03	0.33 ± 0.01	2.84 ± 0.09	0.19 ± 0.01	2.09 ± 0.08
39	1.16 ± 0.04	0.40 ± 0.02	2.97 ± 0.09	0.37 ± 0.03	2.78 ± 0.10
21	1.07 ± 0.05	0.47 ± 0.03	2.36 ± 0.09	0.39 ± 0.04	2.66 ± 0.14
5	0.79 ± 0.10	0.25 ± 0.03	3.12 ± 0.14	0.15 ± 0.03	1.83 ± 0.23
65	1.35 ± 0.02	0.47 ± 3E ⁻³	2.85 ± 0.04	0.49 ± 0.01	3.26 ± 0.03
38	1.00 ± 0.02	0.34 ± 0.01	3.06 ± 0.10	0.26 ± 0.01	2.36 ± 0.04
30	0.98 ± 0.03	0.35 ± 0.01	2.79 ± 0.08	0.25 ± 0.01	2.33 ± 0.06
84	1.10 ± 0.01	0.42 ± 4E ⁻³	2.63 ± 0.03	0.36 ± 0.01	2.71 ± 0.03
49	0.77 ± 0.02	0.32 ± 0.01	2.44 ± 0.06	0.17 ± 0.01	1.84 ± 0.05
6	1.75 ± 0.13	0.62 ± 0.03	2.86 ± 0.28	0.86 ± 0.08	4.27 ± 0.26
46	0.84 ± 0.02	0.41 ± 0.01	2.11 ± 0.06	0.24 ± 0.01	2.07 ± 0.06
4	0.79 ± 0.08	0.33 ± 0.02	2.49 ± 0.37	0.19 ± 0.02	1.94 ± 0.16
28	0.84 ± 0.04	0.35 ± 0.02	2.51 ± 0.10	0.22 ± 0.02	2.05 ± 0.10
9	0.80 ± 0.05	0.31 ± 0.02	2.63 ± 0.19	0.15 ± 0.02	1.75 ± 0.11
16	0.84 ± 0.05	0.23 ± 0.01	3.75 ± 0.20	0.12 ± 0.01	1.80 ± 0.10

and measurements (Average \pm Standard Error). Values are expressed in mm. $P. = P$

*'aleoovoidus*

Table 2. Summary table of values of the Principal Components (PC) of the Fourier co

	n	Mean	S.E.	Min	Max	Median
PC 1	25	62.87	2.39	43.33	98.81	60.06
PC 2	25	22.71	1.62	0.73	36.32	23.8
PC 3	25	7.3	0.69	0.46	14.86	6.89

efficients.

Appendix C

Associate Editor: Dr. Jeffrey Thompson
Subject Editor: Kevin Padian
Editorial Coordinator: Lianne Parkhouse
Royal Society Open Science

Dear Editors

Thank you for considering our manuscript “GEOMETRIC MORPHOMETRICS OF ENDOPHYTIC OVIPOSITION TRACES OF ODONATA (EOCENE, ARGENTINA)” for Royal Society Open Science, and for the opportunity to submit a revised version. We are grateful for all the constructive comments and recommendations from the editor and reviewers, which helped us to improve the quality of the article.

We have made corrections taking into consideration all the comments received. Language and tables were carefully revised to improve the clarity of the presentation.

Below please find our point-by-point responses to the comments received and the changes we have made in the manuscript. The changes were also highlighted in yellow in the manuscript to facilitate the review.

If you need any further clarifications or changes please do not hesitate to contact us.

Looking forward to your response,

Best regards,

Dr. Eugenia Romero-Lebrón (on behalf of all coauthors)

Associate Editor Comments to Author (Dr Jeffrey Thompson):

Dear Eugenia et al,

Your manuscript has been reviewed by three expert reviewers, and I am happy to say that they have all been very positive about your manuscript. I am thus recommending acceptance with minor revisions. There are a few cases where things are not clear in the manuscript, where things are misspelled, or where the English could be improved, which has been highlighted by the reviewers in their attached, annotated, files. There are also a number of stylistic errors found in the references. I thus suggest you go through the manuscript with a fine-toothed-comb to correct these errors in spelling, formatting, and clarity prior to resubmission. Please address all suggestions and comments made by the reviewers prior to resubmission.

All the best,
Jeff Thompson

Re: Thank you for your positive words and for the opportunity to revise and resubmit our work.

Reviewer 1:

I included some corrections and comments in the pdf file of the manuscript (attached)

Re: Thank you for your constructive comments. We appreciate your detailed review.

Point-by-point response:

1. Line 15: authors consider these traces sometimes as scars that surrounds the eggs (traces) as in the first part of the abstract and several parts of the main text BUT also as simple eggs as in the end of the abstract and the end of the Discussion section. These are traces and not eggs thus along the manuscript the authors must argued and consider only the correct expression and definition
Re: We appreciate this observation. The manuscript has been completely revised for conciseness.
2. Line 49: confusion: it seems authors indicate the current study, but clearly is not the case
Re: Writing was revised for clarity.
3. Line 52: several times in the manuscript the spelling of this word is not correct: ichnospecies
Re: Done
4. Line 75: to delete this comma
Re: Done

5. Line 87: several times in the manuscript the spelling of this word is not correct: morphometrics
Re: Done
6. Line 164 and 171: surely better using letters
Re: Done
7. Line 174: Romera-Lebrón et al. Please, review the entire manuscript to check this kind of errata
Re: We apologize for missing this typo, now thoroughly revised, which was related to some issues with the reference manager.
8. Line 174: change “analyse” to “analysed”
Re: Done
9. Line 234 to 236: not clear in each case if the authors refer to extant behaviour or past behaviour or both. In any case, the sentence is confuse
Re: We referred to behaviour inferred from the fossil record. The text was revised for clarity. It now reads: “... in the fossil record Lestidae females lay their eggs in specific substrates such as *Daphnogene* leaves (Lauraceae), while Coenagrionidae females do not appear to be selective.” (line 244 to 246)
10. Line 272: change “icnhnotaxa” to “ichnotaxon”
Re: Done
11. Line 272: before the word ichnotaxon
Re: Done
12. Line 272: Change “Rio” to “Río”. Please, revise the entire manuscript to eliminate this kind of errata
Re: Done
13. Line 281: are, not were, because these are fossil ichnotaxa and originally were ovipositions (no fossils)
Re: Done
14. Line 281: Change “presented” to “present”
Re: Done
15. Line 282: Change “one” to “the same”
Re: Done
16. Line 319: Change “current” to “extant”
Re: Done
17. Line 322: Change “interpreted” to “interpret”
Re: Done
18. Line 331: two sentences that must be separated using . or ;
Re: Done
19. Line 334: Change “Patagonia” to “Patagonian”
Re: Done
20. Line 354: “Rodrigues” maybe it is Rodríguez?
Re: We appreciate the observation and agree that the surname Rodriguez is more usual, but in this case, his surname is Rodrigues (with “s”)

21. Line 362 to 379: comments on these 6 first references showing the need of a detailed review of all the references in the list:
- 1- pages separated by different symbols.
 - 2- names of taxa not in italics.
 - 3- incorrect spelling of the name of one author: Petrulevicius to Petrulevičius.
 - 4- incorrect spelling of the word Fossillagerstatten.
 - 5- in ref 6 there is not; after the year of publication...
- Re: The reference list was carefully revised and corrected.
22. Line 548 and 549: S2-S5 not available to review
- Re: We hope they are available in the resubmission.
23. Figure 1: to include in this part of the figure caption both the taxon and the locality (and age).
- Re: Done
24. Figure 2: Change “Paleoovoidus arcuatus” to “*Paleoovoidus arcuatus*” in italics. To revise the entire manuscript
- Re: There was a format change when we logged the files. We hope it is corrected now.
25. Table 1: very confuse. The genus Lomatia is both Fabaceae and Proteaceae in this table. Please, revise carefully the two tables
- Re: The name was written using a lower alignment, thank you for noticing. Lomatia is a Proteaceae. We made the necessary corrections in the table.
26. Table 2: why only one decimal in this measure if the rest (except to 7.3) have two decimals?
- Re: Done
-

Reviewer 2:

The authors have attempted to study the shape variation of the egg traces of Odonata from the Eocene in Argentina, the article sound very original and the use of elliptical analyses of Fourier are indeed interesting, I missed a more sophisticated analysis implemented in the package “Momocs” for Fourier analyses, the use of old implementation is not bad but always can be better.

I have some comments written in the manuscript PDF please take a look all of them mostly are of form, but I have some principal concern regarding the substrate variation, the authors said very explicit that there are no pattern related, but I noticed at least 3 pattern of 3 different substrate in the PCA, from Eucalyptus chubutensis (very clear variation in mostly all the morphospace) and in Malvaceae (very narrowed group of specimens not superposed to the others).

Nevertheless, I like the idea and I’m sure that the article can be accepted after minor revision comments of the PDF and improve the manuscript

Re: We appreciate your positive words and your detailed review. We also appreciate your suggestion and will use the "Momocs" package in our future research.

In relation to the 3 patterns you mention, we have revised/improved our interpretation accordingly. Please see response to point 16 below.

Point-by-point response:

1. Line 29: Change “4” to “four”
Re: Done
2. Line 30: “Odonata” order? or general, odonates
Re: In this sentence we refer to the sexual behaviour of the order Odonata, since it is the only extant order of Odonatoptera and therefore the only order from which we can know precisely their sexual behaviour.
3. Line 80: what is a CCD?
Re: CCD is the charged-coupled device, which is a transistorized light sensor on an integrated circuit. We now state what the acronym stands for
4. Line 85: better modify to traditional morphometrics is more accurate than classical
Re: We agree
5. Line 151: This information belong to the Methods
Re: Done
6. Line 171: decide 1 or one
Re: One. Done
7. Line 182, 183 and 187: not clear what is wider? Greater?
Re: This paragraph was revised for clarity.
8. Line 183: “(p < 0.05)” which test provide this significance?
Re: Comparisons were made using Generalized and Mixed Linear Models (MLGM) and LSD Fisher as a *posteriori* test (Alpha = 0.05). This is described in the Materials and Methods section (line 109 and 140). We have revised the paragraph for clarity.
9. Line 189: what is relatively little?
Re: By “little variations” we meant that the variations observed are not statistically significant; this is now clearly stated.
10. Line 193: you performed 25 PCAs?
Re: Yes, for each individual substrate we performed a principal component analysis. We have submitted a figure for each material with the contours of each trace and the PCAs that I hope will be available in the electronic version. If not, appendices S1-S5 show the PCAs of 5 materials (MPEF-IC-1382, MPEF-IC-1376, MPEF-IC-1385 and MPEF-IC-1388).
11. Line 194: The figure 4 not correspond to a PCA, correspond to the shape variation, please modify.
Re: Yes, excuse the mixup. We appreciate the observation and have corrected the text accordingly.

12. Line 195: I dont understand this, how a single PC can explain a range of % of shape variation, here the author need to be more specific...
Re: We realize the text was confusing and the paragraph was rewritten. We did not imply that a single PC explained a range of % variation; instead, we referred to the range of variation explained when considering the 25 PCA we carried out (one for each substrate). We have revised the text for clarity. We also replaced Table 2, which was meant as an overall summary, with the PCA for each of the 25 substrates.
 13. Line 196: not clear what are you saying here.
Re: Component 1 mainly discriminated the shape of the apex of the trace. The text was rewritten.
 14. Line 204: Change “total variation observed” to “total shape variation”
Re: Done.
 15. Line 211: Change “in relation to” to “related to”
Re: Done
 16. Line 216: Im disagree with the authors in this observation, Malvaceae is indeed very different to the other groups, and the *Eucalyptus chubutensis* and *Myrcia deltodea* also have some important tendency
Re: We agree, you are right and we appreciate this observation. We have improved the description of the figure in the text and its discussion. It is true that Malvaceae is different from other substrates, but we prefer to take this result very carefully. To draw robust conclusions from this family is complicated as there is only 1 material which has only 5 traces (Table 1) (please see the paragraph starting on line 223 of the results section and then line 336 of the discussion section).
 17. Line 221: reduced the morphospace to only 3 PC's? not clear what is reduced here...
Re: The text was corrected. The first three components together explain nearly 93% of shape variability.
 18. Line 321: At least Malvaceae was very clear, please explain why this substrate could be different to the others... Also explain why the shape variance of *Eucalyptus chubutensis* is higher than all the other substrate
Re: As we explained in point 16 you are right to note the pattern in the Malvaceae family, and this is now mentioned in the manuscript. But because of the N (1 specimen with only 5 traces) we chose to be careful in drawing conclusions about the Malvaceae family. On the other hand, *Eucalyptus chubutensis* showed the widest morphospace, which in part could be explained by the larger number of separate samples (N = 4) analysed (see Figure 6 B in revised version). However, *Myrcia deltodea* also shows a wide morphospace, with all traces coming from the same substrate material. Therefore, we interpret that the substrate would not be exerting differential shape modification pressures on the egg traces between the studied materials.
-

Reviewer 3:

Dear Authors,

Congrats for your manuscript. There is tiny mistakes to be corrected with english and one sentence in the discussion that seems awkward to me.

Otherwise, manuscript deserves to be publish as is.

Illustration and the effort of represent each oviposition per leaf in a draw are very good.

Re: Thank you for your positive words and constructive comments.

Point-by-point response:

1. Line 171: I don't understand the end of this sentence. What do you mean by "unidentified" followed by "now assigned to" ?

Re: Text was revised for clarity; we meant that we assigned the specimen to the ichnoclass.

2. Line 284 and 296: Change “ichnospecie” to “ichnospecies”

Re: Done

3. Line 339: I do not appreciate this sentence despite the fact I understand what do you mean. It look akward to write "as far as we know" in a scientific publication or not. This sentence looks like to say: according our personal memory, there is no study. No, you should demonstrate in this sentence that you search on the litterature and there is nothing. ☺

Re: We appreciate the observation. We modified the text to make it clear that we searched and did not find any literature on the subject. ☺